# Effects of variable, ice-ocean surface properties and air mass transformation on the Arctic radiative energy budget

Manfred Wendisch[1], Johannes Stapf[1], Sebastian Becker[1], André Ehrlich[1], Evelyn Jäkel[1], Marcus Klingebiel[1], Christof Lüpkes[2], Michael Schäfer[1], and Matthew D. Shupe[3,4]

[1]Leipzig Institute for Meteorology (LIM), Leipzig University, Germany
[2]Alfred Wegener Institute (AWI), Helmholtz Center for Polar and Marine Research, Bremerhaven, Germany
[3]Physical Sciences Laboratory (PSL), National Oceanic and Atmospheric Administration (NOAA), Boulder, Colorado, USA
[4]Cooperative Institute for Research in Environmental Sciences (CIRES), University of Colorado, Boulder, USA

**Correspondence:** Manfred Wendisch (m.wendisch@uni-leipzig.de)

**Abstract.** Low-level airborne observations of the Arctic surface radiative energy budget are discussed. We focus on the terrestrial part of the budget, quantified by the thermal-infrared net irradiance (TNI). The data have been collected in cloudy and cloud-free conditions over and in the vicinity of the marginal sea ice zone (MIZ) close to Svalbard during two aircraft campaigns conducted in spring of 2019 and in early summer of 2017. The measurements, complemented by ground-based observations available from the literature and radiative transfer simulations, are used to evaluate the influence of surface type (sea ice, open ocean, MIZ), seasonal characteristics, and synoptically driven meridional air mass transports into and out of the Arctic on the near-surface TNI. The analysis reveals a typical four-mode structure of the frequency distribution of the TNI as a function of surface albedo, sea ice fraction, and surface brightness temperature. Two modes prevail over sea ice and another two over open ocean, each representing cloud-free and cloudy radiative states. Characteristic shifts and modifications of the TNI modes during the transition from winter towards early spring and summer conditions are discussed. Furthermore, the influence of warm air intrusions (WAIs) and marine cold air outbreaks (MCAOs) on the near-surface downward thermal-infrared irradiances and the TNI is highlighted for several case studies. It is concluded that during WAIs the surface warming depends on cloud properties and evolution. Lifted clouds embedded in warmer air masses over a colder sea ice surface, decoupled from the ground by a surface-based temperature inversion, have the potential to warm the surface more strongly than near-surface fog or thin low-level boundary layer clouds, because of a higher cloud base temperature. For MCAOs it is found that the thermodynamic profile of the southward moving air mass adapts only slowly to the warmer ocean surface.

## 1 Introduction

During the last decades, the Arctic climate has experienced significant transformations that are still ongoing (Jeffries et al., 2013; Koenigk et al., 2020). They were triggered by global warming pushing local and remote feedback mechanisms, which function more efficiently in the Arctic than anywhere else on Earth (Goosse et al., 2018; Wendisch et al., 2023). One of the most obvious current signs of the changing Arctic climate is the enhanced warming of the Arctic relative to the entire globe (Thoman et al., 2020; IPCC, 2021), which appears two to four times stronger than the global warming (Rantanen et al., 2022).

As another indication of the ongoing changes of the Arctic climate system, the Arctic sea ice extent has halved within the last 40 years (Meier et al., 2014; Stroeve and Notz, 2018; Screen, 2021). These and other rather dramatic and rapid Arctic climate changes are caused by specific Arctic processes and feedback mechanisms that lead to an overall higher sensitivity of the Arctic climate system to global warming (Wendisch et al., 2023). This enhanced efficiency of various coupled mechanisms within the

Arctic climate system is commonly referred to as Arctic amplification (Serreze and Barry, 2011).

The two phenomena mentioned above (amplified Arctic warming, sea ice retreat) are of eminent importance for the Arctic surface radiative energy budget (Pithan and Mauritsen, 2014; Block et al., 2020). Besides the solar component that is not considered in this paper, the surface radiative energy budget is quantified by the thermal-infrared ($\sim 3$–$50\,\mu m$) net irradiance (TNI[1]), $F_{\mathrm{tir,net}}$. The TNI is given by the difference of downward ($F_{\mathrm{tir}}^{\downarrow}$) and upward ($F_{\mathrm{tir}}^{\uparrow}$) thermal-infrared irradiances:

$$F_{\mathrm{tir,net}} = F_{\mathrm{tir}}^{\downarrow} - F_{\mathrm{tir}}^{\uparrow}. \tag{1}$$

Because of its importance, the TNI needs to be represented realistically in climate models, in order to describe and disentangle the concurrent feedback processes within the Arctic climate system. However, in particular in the Arctic, the TNI derived from global reanalysis (Graham et al., 2019) and regional climate models (Pithan et al., 2014; Sedlar et al., 2020) entails serious deficiencies. To address these issues, this paper explores airborne *in situ* observations of the TNI close to the Arctic surface

that are available for comparisons with models.

The few ground-based and long-term measurements of the Arctic TNI were mostly performed at locally fixed measurement stations over homogeneous, highly reflective surface (sea ice, snow). These observations over sea ice revealed two characteristic radiative winter states driven by the absence or presence of optically thick clouds (Stramler et al., 2011; Graham et al., 2017). Airborne measurements showed that the seasonal changes of surface characteristics and atmospheric thermodynamic structures

influence these two common radiative states of the Arctic TNI (Stapf et al., 2021a; Stapf, 2021). These observations are particularly well suited to study the influence of surface type on the mode structure of TNI by enabling collection of data over spatially variable surfaces. This is especially important in the marginal sea ice zone (MIZ) where sea ice, cloud, and thermodynamic conditions strongly vary over small horizontal scales. Because of this variability, important processes such as cloud-radiation interactions over different surface types, or thermodynamic adjustment of air masses, also vary strongly over

the MIZ (Gardner and Sharp, 2010; Stapf et al., 2021a).

In addition to local surface conditions and thermodynamic profile properties, such as air temperature, water vapor content, and clouds that are the main drivers of downward atmospheric emission in the Arctic atmospheric boundary layer (ABL), the TNI is also determined by remote influences, such as large-scale circulation patterns that are associated, for example, with marine cold air outbreaks (MCAOs) or warm air intrusions (WAIs) (Pithan et al., 2018). The North Atlantic is characterized

by a high frequency of occurrence of MCAOs, which are an important driver of strong heat losses from the ocean and the generation of dense ocean waters relevant for ocean circulations (Papritz and Spengler, 2017). MCAOs have been studied extensively using airborne *in situ* (Brümmer, 1996, 1997) and ground-based (Geerts et al., 2022) observations, which showed highly uncertain contributions of radiative energy fluxes to the ABL energy budget. In modeling studies of MCAOs, radiative

---

[1]All acronyms are listed and explained in the Appendix

processes received less attention in the discussion of ABL evolution (Chechin and Lüpkes, 2017) or heat balance (Papritz and Spengler, 2017). The North Atlantic also represents one of the major transport pathways of warm and humid air masses into the Arctic (Mewes and Jacobi, 2019), so-called WAIs. Air mass transports during WAIs modify the local near-surface TNI (Binder et al., 2017; Yamanouchi, 2019). They can trigger large scale melt events over sea ice in summer (Tjernström et al., 2015) and impact sea ice evolution in winter (Persson et al., 2017). Conversely, the thermodynamic transformation along an air mass trajectory can be influenced by the near-surface TNI. In this way, such air mass transformations feed back into the TNI by changing cloud, atmosphere, and surface properties. Therefore, a characterization of the transformation process is key to understanding the consequences for the local surface radiative energy budget (Tjernström et al., 2019). Unfortunately, these synoptically driven transformation processes during MCAOs and WAIs are not well captured in models predicting the future of the Arctic climate (Pithan and Mauritsen, 2014).

Previous airborne campaigns examined the North Atlantic-Arctic domain, in particular the Fram Strait, to study turbulent, radiative, and cloud processes (Hartmann et al., 1992; Brümmer, 1996; Wendisch et al., 2019). Furthermore, ABL processes have been investigated (Lampert et al., 2012; Vihma et al., 2014; Tetzlaff et al., 2015), and air mass transformations were observed during MCAOs (Brümmer, 1996; Chechin et al., 2013). In contrast to these former investigations, our study focuses on the near-surface TNI in the heterogeneous region of the MIZ and over adjacent sea ice-covered and open ocean areas. Low-level airborne observations of TNI collected during two aircraft campaigns characterized by spring and early summer conditions are presented and discussed in terms of the observed mode structure of the near-surface TNI. Furthermore, several case studies of WAIs and MCAOs crossing the MIZ are presented and related radiative and adjustment effects are investigated.

After an introduction of the observations including the instrumentation and the applied radiative transfer model in Section 2, a summary of the observed surface and atmospheric conditions prevailing during the two discussed airborne campaigns is given in Section 3. The typical mode structures of the TNI in cloudy and cloud-free conditions and their links to surface type and atmospheric thermodynamic profiles is investigated in comparison to other available data in Section 4. Section 5 presents case studies of the impact of WAIs and MCAOs on downward thermal-infrared irradiance close to the ground, and on the derived TNI. Furthermore, the possible radiative contributions to associated ABL transformations are discussed. Finally, Section 6 summarizes and concludes the paper.

## 2 Measurements and radiative transfer simulations

The data discussed here were collected northwest of Svalbard during the AFLUX (Airborne measurements of radiative and turbulent FLUXes of energy and momentum in the Arctic ABL) and ACLOUD (Arctic CLoud Observations Using airborne measurements during polar Day) campaigns. The flight tracks and mean sea ice conditions during the campaigns are illustrated in Fig. 1. The airborne observations were conducted over sea ice and open ocean in the vicinity of the MIZ. While AFLUX was conducted in spring (19 March to 11 April 2019), ACLOUD was performed in early summer (23 May to 26 June 2017). During AFLUX, the sea ice edge was situated slightly further to the North compared to ACLOUD. The scientific background and first results of ACLOUD are introduced by Wendisch et al. (2019), the synoptic overview is given by Knudsen et al. (2018).

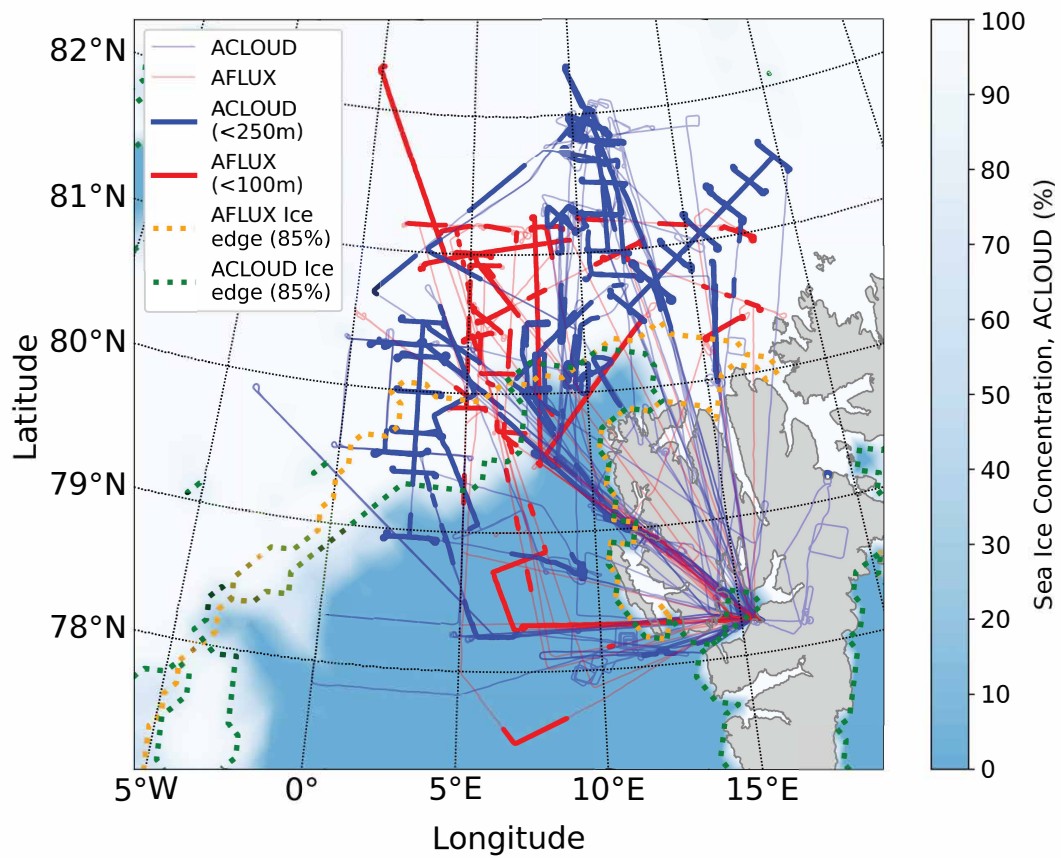

**Figure 1.** Flight paths during ACLOUD (thin blueish lines) and AFLUX (thin reddish). Only data measured during low-level flight sections are analyzed, which are indicated by thick blue (ACLOUD) and thick red (AFLUX) lines, respectively. Low-level is defined as flight altitudes less than 250 m for ACLOUD and less than 100 m for AFLUX. The sea ice concentration during ACLOUD is indicated by the color bar, in addition the sea ice edges (defined as the 15 % isoline of satellite-derived sea ice fraction) observed during AFLUX and ACLOUD are depicted as dotted orange (AFLUX) and dotted green (ACLOUD) lines, respectively.

The scientific instrumentation and the data processing methods are presented for ACLOUD by Ehrlich et al. (2019b) and for AFLUX by Mech et al. (2022), respectively. While during ACLOUD the two research aircraft Polar 5 and Polar 6 of the Alfred Wegener Institute, Helmholtz Centre for Polar and Marine Research (AWI) were employed (Wesche et al., 2016), only Polar 5 was used during AFLUX. Polar 5 was equipped during AFLUX similarly compared to ACLOUD.

5    In this paper, data obtained during low-level flight sections are analyzed, either below clouds or in cloud-free conditions. During AFLUX, six hours (corresponding to a horizontal distance of 1400 km) of low-level data were gathered at an altitude below 100 m (average 73 m) (Stapf et al., 2021b). During ACLOUD, 16 hours of measurements at an altitude of less than 250 m (average 80 m), covering a total distance of 3700 km, were collected (Stapf et al., 2019). The corresponding low-level flight sections are indicated by thick lines in Fig. 1.

Broadband pyrgeometers (type CGR4, see https://www.kippzonen.com/, wavelength range of 4.5–42 μm) measured the upward and downward thermal-infrared irradiances to derive the near-surface TNI, see Eq. (1). In addition, upward and downward solar irradiances were measured using pyranometers (type CMP22, see https://www.kippzonen.com/, wavelength range of 0.2–3.6 μm) to derive the surface albedo (ratio of upward and downward solar irradiances). On both aircraft (Polar 5 and Polar 6), pairs of upward and downward looking pyrgeometers and pyranometers were installed. The irradiance data were recorded with a frequency of 20 Hz. Furthermore, a 180° downward-facing fish-eye camera was installed to determine the cosine-weighted sea ice fraction ($I_f$) with a domain size depending on flight altitude (Jäkel et al., 2019). A downward-looking Kelvin infrared thermometer (KT-19, version KT19.85 II) provided the surface brightness temperature (BT). The KT-19 collects nadir radiance with a field of view of 2° (Ehrlich and Wendisch, 2015). These measurements were converted into surface skin temperature values assuming an emissivity of 1. This is justified due to the small impact of atmospheric absorption in the wavelength range of 9.6–11.5 μm for which the KT-19 is sensitive (Ehrlich et al., 2019b). The local atmospheric thermodynamic state and basic meteorological parameters were derived from dropsondes and aircraft *in situ* measurements during ascents and descents in the vicinity of the low-level flight sections. During AFLUX and ACLOUD, radiosonde data from the permanent AWIPEV research base at Ny-Ålesund (Svalbard) were used. During ACLOUD, additional radiosondes were launched from the Research Vessel (RV) *Polarstern* in the framework of the concurrent PASCAL (Physical Feedbacks of Arctic Layer, Sea Ice, Cloud and Aerosol) campaign (Wendisch et al., 2019).

Due to the generally colder in-cloud temperature observed during AFLUX, icing of the instruments was more likely during this campaign compared to ACLOUD. Especially during longer horizontal flight sections in super-cooled cloud top regions, occasionally sudden and persisting ice caps occurred on the glass domes of the pyranometers. In cloudy conditions with mainly diffuse solar radiation, icing is difficult to detect and remains a possible source of erroneous measurements of solar and thermal-infrared irradiances (Cox et al., 2021). In total, using a combination of radiative transfer simulations and manual tests of plausibility, about 21 % of the solar irradiance measurements during AFLUX were flagged as being influenced by icing. These invalid data were discarded from further analysis. Due to the flat dome of the pyrgeometer, the terrestrial irradiances were less affected by severe icing. In total, about 5 % of the thermal-infrared pyrgeometer data collected during low-level flight sections had to be discarded.

The cloud liquid water path (LWP) was derived during below-cloud, low-level flights applying a transmissivity-based retrieval technique (Stapf et al., 2020). The obtained values represent an equivalent LWP (assuming a cloud droplet effective radius of 8 μm), as cloud ice has not been considered in the retrieval. For large values of solar zenith angles (SZA), significantly higher uncertainties of the retrieved LWP have to be considered, because longer horizontal path lengths are involved in the retrieval. Furthermore, three-dimensional radiative effects of heterogeneous mid-level cloud fields may occasionally introduce unreasonably high LWP values (larger than $200 \, \mathrm{g \, m^{-2}}$), which were excluded from the analysis. The solar irradiance measurements in cloud-free conditions are seriously distorted at high values of SZA common in the Arctic, because of deviations from the horizontal alignment of the pyranometers, even if geometrically corrected (Wendisch et al., 2001). Therefore, a threshold for the angular deviations from the horizontal orientation of the pyranometers was defined (5° for AFLUX, 4° for

ACLOUD) that needed not to be exceeded to qualify a measurement as valid. This requirement led to a reduction of the data set of solar irradiances by 28 %.

Besides the observations, radiative transfer simulations were performed to reproduce the airborne irradiance measurements. For this purpose, the *libRadtran* package (Emde et al., 2016) was employed in a setup described by Stapf et al. (2020). The local thermodynamic profile measurements obtained from dropsondes released during the flights or *in situ* profile data collected by the sensors mounted on the aircraft were merged with nearby upper air radiosonde observations.

The airborne observations provide the TNI and albedo at flight level close to the ground. These data may slightly differ from the corresponding surface values depending on the atmospheric aerosol and thermodynamic conditions even in case of flight altitudes less than $100\,\mathrm{m}$ (Wendisch et al., 2004). Especially over open ocean and leads, occasionally large differences between the air temperature at $60\,\mathrm{m}$ flight altitude and the surface temperature of up to $20\,^\circ\mathrm{C}$ may occur. For such extreme cases, radiative transfer simulations have indicated a difference between cloud-free TNI observed at a flight altitude of $60\,\mathrm{m}$ and the ground of up to $10\,\mathrm{W\,m^{-2}}$. For strong surface temperature inversions over sea ice, up to $2\,\mathrm{W\,m^{-2}}$ difference were simulated for the TNI. During AFLUX, the visibility during low-level flight sections was often reduced. Over leads and open ocean, sea smoke developed. Furthermore, in the low ABL over sea ice occasional surface-based clouds or precipitation and also fog were observed. These circumstances may slightly bias the thermal-infrared and solar irradiance measurements and the derived parameters (TNI and surface albedo) observed at flight altitude, but they were not excluded.

## 3 Surface and atmospheric conditions during AFLUX and ACLOUD

### 3.1 Surface and cloud properties

Figure 2 depicts the frequency distributions (in the form of probability density functions, PDFs) of the surface albedo, the surface BT, and the equivalent LWP as a function of the sea ice fraction $I_\mathrm{f}$, separated for the AFLUX and ACLOUD campaigns. The surface albedo observations (Figs. 2a and 2b) indicate that the majority of the low-level flights were conducted either over sea ice with concentrations higher than 80 %, or over mostly open ocean with sea ice concentrations lower than 20 %. During ACLOUD a higher fraction of flights were conducted over fractional sea ice with variable concentrations between 50 % and 95 %. The surface albedo and sea ice fraction are roughly linearly correlated with broader distributions for ACLOUD during early summer when melt ponds start to develop. The sea ice fraction modifies the value of the surface albedo as a weighted average of open ocean and sea ice albedo and the three-dimensional multiple reflections between atmosphere and surface adjust the value. It is interesting to note that for a given sea ice fraction the maxima of the PDFs of surface albedo during AFLUX are larger compared to ACLOUD. The reason is that AFLUX was conducted earlier in the year compared to ACLOUD, and thus, during AFLUX larger values of SZA of $72^\circ$ to $82^\circ$ prevailed, compared to $55^\circ$ to $69^\circ$ during ACLOUD. For cloud-free conditions, the surface albedo is larger in case of larger values of SZA. Also the colder and fresher snow conditions with smaller snow grain sizes contribute to the larger albedo values over sea ice during AFLUX. There is some hint that the frequency distribution of surface albedo obtained from measurements during ACLOUD over homogeneous sea ice actually indicates two slight sub-modes. They reflect somewhat higher surface albedo values before the beginning of the melt season

and slightly decreasing surface albedos at the end of the campaign during early summer. Furthermore, the surface albedo of the open ocean is influenced by the different SZAs during ACLOUD and AFLUX, which explains the broader mode observed for AFLUX from 0.05 (cloudy) to 0.3 (cloud-free) compared to the rather narrow mode from 0.05 to 0.15 for ACLOUD. Over open ocean, the sea smoke mentioned earlier might have increased the value of reflected irradiance, producing an apparent increase of the surface albedo, especially in cloudy conditions.

The characteristics of the different seasons are evident in the surface BT distributions (Figs. 2c and 2d). During AFLUX, low surface temperatures of often below $-20\,^\circ$C and surface temperature gradients in the MIZ of up to 25 $^\circ$C were detected. The data are affected by a pronounced daily variability depending on the synoptic situation and distribution of clouds. Also more complex surface types like nilas (covered or uncovered by thin snow) contributed to the broadening of the surface temperature distribution during AFLUX. During ACLOUD, after an intermediate cold period at the end of May, a rather uniform surface temperature was observed in the early melt season in the MIZ, with surface temperatures distributed around the melting point of snow and the freezing point of sea water.

In Figures 2e and 2f, the PDFs of the retrieved equivalent LWP illustrate that during ACLOUD cloud fields with an LWP larger than $30\,\mathrm{g\,m^{-2}}$ are more frequently observed than during AFLUX. The median LWP in cloudy conditions above sea ice (LWP > $5\,\mathrm{g\,m^{-2}}$, $I_\mathrm{f}$ > 95 %) amounts to $34\,\mathrm{g\,m^{-2}}$ for AFLUX and $50\,\mathrm{g\,m^{-2}}$ for ACLOUD. In 80 % of the conditions, the LWP was lower than $58\,\mathrm{g\,m^{-2}}$ during AFLUX and lower than $68\,\mathrm{g\,m^{-2}}$ during ACLOUD. During the low-level flights of AFLUX, thin and low cloud fields were often observed in a shallow ABL over sea ice. Sea smoke and surface-based clouds were identified over open leads embedded in the sea ice or during MCAOs above the open ocean. Over sea ice, shallow fog-like conditions were observed more frequently during AFLUX, compared to ACLOUD, where mostly a clearly separated low cloud base was present with occasional precipitation. During the low-level flight sections of ACLOUD, more homogeneous cloud fields in an ABL with larger vertical extent prevailed.

## 3.2 Vertical temperature structure and stability of the lower atmosphere

The average temperature profiles measured during AFLUX and ACLOUD over homogeneous sea ice and open ocean areas are depicted in Figs. 3a and 3b. During AFLUX (spring), a strong temperature inversion above a shallow and cold ABL was frequently observed over sea ice (Fig. 3a). The near-surface lapse rate of the individual profiles depended more on the prevailing conditions over sea ice. They quickly adapted to the cloud conditions and transitioned between strong surface-based inversions in cloud-free conditions and more neutral profiles in cloudy situations (Stapf et al., 2021a). During ACLOUD (early summer), the average temperature profile measured over sea ice is shaped by frequently observed ABL clouds with a cloud top temperature inversion at altitudes between $0.3\,\mathrm{km}$ and $0.4\,\mathrm{km}$ (Fig. 3b). Compared to the average temperature profiles measured over the sea ice, the profiles over the open ocean depict a significant increase of both the ABL height and temperature during both AFLUX and ACLOUD. The vertical thermodynamic structure over sea ice transforms from a surface inversion-dominated, often cloud-free spring atmosphere towards a more cloud-dominated ABL in early summer.

Figure 3c illustrates the PDF of the difference of potential air temperatures ($\Delta\theta = \theta_\mathrm{atm} - \theta_\mathrm{sur}$) measured at low-level flight altitudes ($\theta_\mathrm{atm}$) and at the surface ($\theta_\mathrm{sur}$) for the two campaigns. The surface temperature is assumed to be the skin temperature

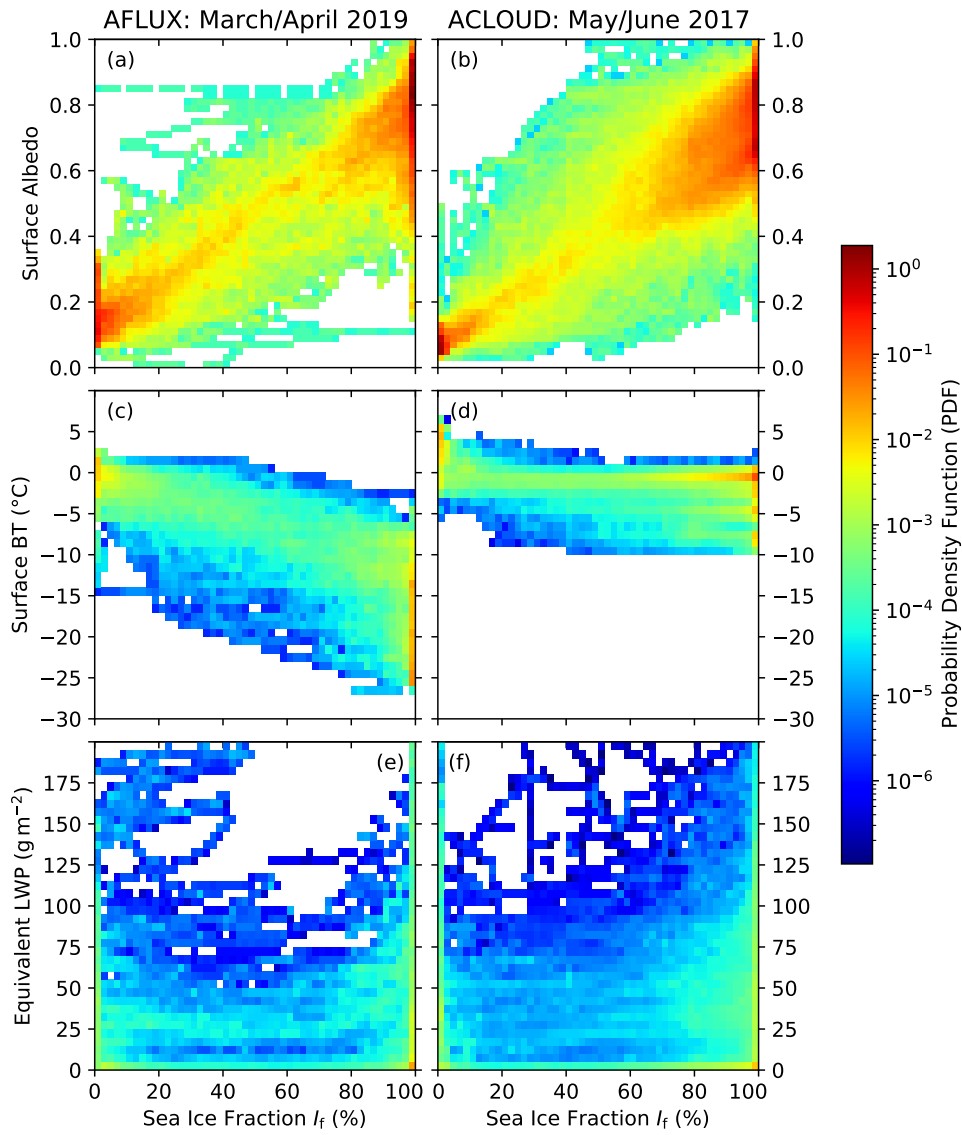

**Figure 2.** Histograms of probability density functions of observed and derived surface (albedo, brightness temperature) and cloud (Liquid Water Path, LWP) properties as a function of cosine-weighted sea ice fraction ($I_f$). (a) and (b): surface albedo; (c) and (d) surface brightness temperature (BT); (e) and (f): equivalent cloud liquid water path (LWP).

derived from the KT-19. Positive values of $\Delta\theta$ indicate stable thermodynamic situations, whereas negative values characterize unstable conditions. Over homogeneous sea ice (red), the surface was often slightly warmer compared to the air above, with mode values of the PDFs of $\Delta\theta$ between $-1\,^\circ$C (AFLUX) and $-3\,^\circ$C (ACLOUD). These values represent mostly cloudy conditions with an almost neutral to slightly unstable near-surface layer. The occurrence of thin nilas and likely fluctuations of

ice floe thickness occasionally caused strongly unstable near-surface layers even over sea ice, in particular during ACLOUD with $\Delta\theta$ values of more than $-10\,^\circ$C to almost $-15\,^\circ$C were observed. During ACLOUD, a second PDF mode around $3\,^\circ$C indicates the occurrence of a significant number of cases with stable atmospheric stratification over sea ice, which mostly appears in cloud-free conditions with strong surface-based inversions. However, over open ocean (blue) and during off-ice flows, cold air masses were advected above the warm open ocean surface causing strongly unstable conditions with strong

convection during both AFLUX and ACLOUD. During ACLOUD, $\Delta\theta$ values over open ocean were less extreme compared to AFLUX. When less cold air is transported southward or warm air is advected northward, moderate negative values of $\Delta\theta$ and even positive spreads (stable layering) were occasionally observed. These cases were often combined with strong surface-based temperature inversions above the open ocean.

## 4    Cloudy and cloud-free modes of thermal-infrared net irradiance (TNI)

In the following section we look at the structure of the thermal-infrared irradiance field measured close to the ground from a statistical point of view. We discuss modes of the frequency distributions of TNI observed during different seasons and over various surfaces.

### 4.1    Seasonal characteristics of the two-mode structure of TNI observed over homogeneous sea ice

To put the observations from AFLUX and ACLOUD into a seasonal context, data from the Surface Heat Budget of the Arctic

Ocean (SHEBA) campaign performed in the Beaufort and Chukchi Seas (Uttal et al., 2002), and the Norwegian young sea ice experiment (N-ICE2015) conducted over sea ice and the MIZ north of Svalbard (Walden et al., 2017) are additionally included in our analysis. Figure 4 shows the frequency distributions of the TNI observed over a homogeneous sea ice surface during SHEBA, N-ICE2015, AFLUX, and ACLOUD in winter (Fig. 4a), spring (Fig. 4b), and early summer (Fig. 4c). From these plots, the common and well-known two-mode structure (typical two maxima) of the TNI field appears (Stramler et al., 2011;

Graham et al., 2017). One mode (called the cloudy mode) is comprised of TNI values centered around zero watts per meter squared, wherein optically opaque low-level clouds are in near radiative balance with the surface. Occasionally, this cloudy mode is slightly shifted to negative TNI values. The second, so-called cloud-free mode typically appears between $-30\,\mathrm{W\,m^{-2}}$ and $-80\,\mathrm{W\,m^{-2}}$. These negative TNI values indicate a net radiative energy loss at the surface, which is characteristic for cloud-free conditions wherein the downward atmospheric emission is smaller than the upward emission from the relatively warmer

surface. The distinction between these two radiative states is typically driven by the presence or absence of super-cooled liquid water clouds, although the cloudy mode can include very thick ice clouds, while the cloud-free mode might also include some radiatively transparent ice cloud layers (Shupe and Intrieri, 2004).

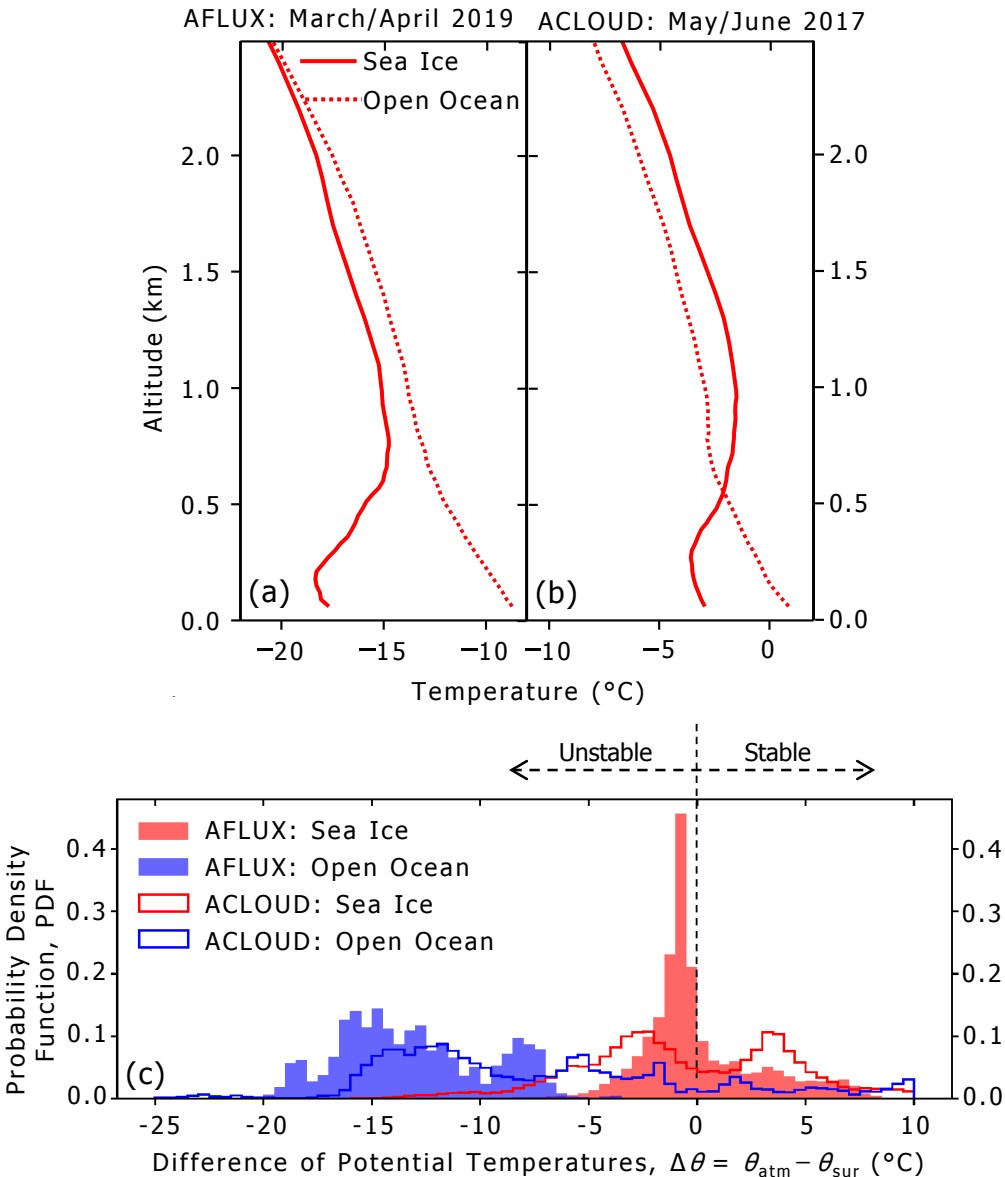

**Figure 3.** (a) and (b): Seasonal, daily averaged temperature profiles observed during AFLUX and ACLOUD above sea ice (solid lines) and open ocean (dotted lines, all-sky conditions). Sea ice areas are defined as regions with a sea ice concentration above 90 % derived from daily sea ice concentrations maps from Spreen et al. (2008). (c) Probability density functions (PDFs) of the difference between potential air temperatures measured at low-level flight altitudes ($\theta_{\mathrm{atm}}$) and at the surface ($\theta_{\mathrm{sur}}$).

The data depicted in Fig. 4 show that the median TNI values (Table 1) for the two modes depend on season. Due to the seasonal cycle of the thermodynamic profiles, the cloud-free mode is shifted from median values of about –40 W m$^{-2}$ during winter (SHEBA), to roughly –60 W m$^{-2}$ during spring AFLUX), and eventually to about –75 W m$^{-2}$ during early summer (ACLOUD). For the cloudy mode a slight trend from almost balanced values to slightly negative TNIs from winter towards
early summer is observed with median values decreasing from –2 W m$^{-2}$ (SHEBA in winter), to –6 W m$^{-2}$ (AFLUX in spring), and –11 W m$^{-2}$ (ACLOUD in early summer). The cloud-free mode detected during AFLUX (Fig. 4b) includes extremely negative TNIs with values sometimes less than –85 W m$^{-2}$, although they occur quite rarely. These strongly negative TNI values result from sporadically high surface temperature, caused by snow-covered thin nilas or thin ice floes evolving during spring. These spots of significantly warmer surface temperatures boost the upward thermal-infrared irradiances emitted
by the surface locally. During ACLOUD (Fig. 4c), surface temperature was rather uniform and warm around the melting point during the beginning of melt season, causing strongly negative values of TNIs as well, which are distributed over a broad range.

| Season (Campaign) | Cloudy | Cloud-free |
|---|---|---|
| Winter (SHEBA) | -2 | -40 |
| Spring (AFLUX) | -6 | -60 |
| Early Summer (ACLOUD) | -11 | -75 |

**Table 1.** Examples of median values (in units of W m$^{-2}$) of frequency distributions of thermal-infrared net irradiances ($F_{\mathrm{tir,net}}$) derived from three campaigns (SHEBA, AFLUX, ACLOUD), which were performed during different seasons (see Fig. 4).

The stronger negative TNI values in summer compared to winter are partly caused by the non-linearity of the Planck emission. In cloud-free conditions, the same temperature difference between surface and atmosphere ($\Delta T = T_{\mathrm{sur}} - T_{\mathrm{atm}}$) results
in different TNI values in winter and summer. To approximately quantify this effect, we estimate the TNI in cloud-free conditions, based on Eq. (1) and using the black-body Stefan-Boltzmann equation for the downward and upward thermal-infrared irradiances:

$$F_{\mathrm{tir,cf}}^{\downarrow} = \sigma \cdot T_{\mathrm{atm}}^4 \,, \text{and} \tag{2}$$

$$F_{\mathrm{tir,cf}}^{\uparrow} = \sigma \cdot T_{\mathrm{sur}}^4 \,, \tag{3}$$

with $\sigma$ the Stefan-Boltzmann constant. The Stefan-Boltzmann law is derived by integration of the Planck law. In winter (assumed $T_{\mathrm{sur}} = -20\,°\mathrm{C}$), a temperature difference of $\Delta T = 20\,°\mathrm{C}$ (meaning a 20 °C warmer atmosphere compared to the surface) yields a TNI of –65 W m$^{-2}$, while in summer (assumed $T_{\mathrm{sur}} = 0\,°\mathrm{C}$) the same $\Delta T = 20\,°\mathrm{C}$ yields a stronger cooling with values of TNI in the range of –83 W m$^{-2}$. The result of this simple approximation roughly agrees with the observed seasonal differences of median values of TNI in cloud-free conditions depicted in Fig. 4, which are in the range of a little more than
20 W m$^{-2}$. However, the lapse rate and, thus, the thermodynamic stability in cloud-free conditions decreases from winter to

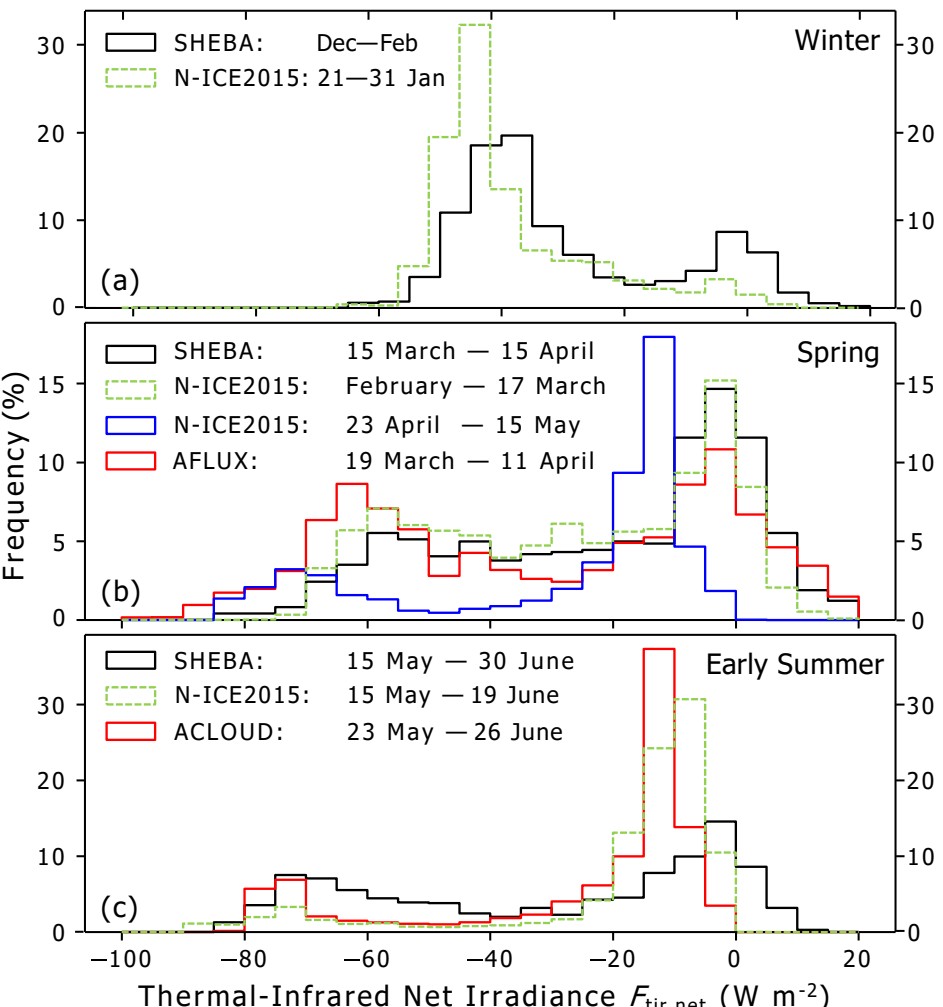

**Figure 4.** Histograms of thermal-infrared net irradiances (TNIs) observed during SHEBA, N-ICE2015, AFLUX, and ACLOUD ($I_f > 90\,\%$) for different seasons: (a) Winter, (b) spring, (c) early summer. No N-ICE2015 observations are available during the AFLUX period, and only a short, mainly cloud-free period in January. The ACLOUD distribution was weighted daily to reduce the impact of one extensive flight in cloud-free conditions.

summer, which lowers the downward thermal-infrared irradiances in cloud-free conditions ($F^{\downarrow}_{\text{tir,cf}} \sim T^4_{\text{atm}}$) in summer. This causes even more negative TNI values in cloud-free conditions in summer.

In cloudy conditions, however, this seasonal shift of the mode values of TNI is diminished, because the difference between effective cloud-base and surface temperatures $\Delta T$ is small compared to the difference between surface and atmosphere tem-
5   peratures in the cloud-free case. Assuming a difference of $\Delta T = 2\,°\text{C}$ between cloud base and surface temperatures results in a TNI of $-7\,\text{W\,m}^{-2}$ and $-9\,\text{W\,m}^{-2}$ for the above mentioned simplified winter and summer scenarios. Thus, for the cloudy

mode only a slight negative shift from winter towards summer can be expected due to the Planck emission non-linearity. During winter (Fig. 4a, SHEBA) and spring (Fig. 4b, N-ICE2015), the lower atmosphere was often stably stratified, promoting a warmer cloud base temperature compared to the surface. Therefore, the corresponding cloudy modes are mostly distributed around zero. The impact of synoptic systems as well as the thermodynamic diversity observed during AFLUX in early spring explain the broadened cloudy modes, which were also observed during SHEBA.

As a consequence of these various processes, the spread between cloud-free and cloudy modes (defined as the difference of median values of modes of TNI frequency distributions) tends to increase from winter towards summer. In Fig. 4, this spread increases from $39\,\mathrm{W\,m^{-2}}$ in winter (SHEBA) to $56\,\mathrm{W\,m^{-2}}$ in spring (AFLUX), and $63\,\mathrm{W\,m^{-2}}$ in summer (ACLOUD). Furthermore, the seasonal cycle of cloud fraction may play a crucial role in the interpretation of Fig. 4. During winter observations of SHEBA (Fig. 4a), the cloud-free conditions outweighed the cloudy state, and the same was true for N-ICE2015 despite a shorter observational record. However, spring conditions are more diverse, with a trend towards more frequent cloudy conditions. In early summer, N-ICE2015 and ACLOUD data indicate a dominant and less variable cloudy mode with only occasional and shorter cloud-free periods, while SHEBA data show relatively more cloud-free conditions (Fig. 4b). With overall less frequent and probably shorter cloud-free periods in early summer (Fig. 4c), the thermodynamic adjustments of the cloud-free atmosphere appear to be less pronounced, resulting in weaker surface-based temperature inversions, which contribute to more negative TNI in the cloud-free states.

With the shift to cloud-dominated synoptic conditions in late spring and early summer, the Arctic ABL becomes geometrically thicker and less stable, which supports colder cloud base temperatures relative to the surface depending on the cloud base height. Over sea ice, slightly negative TNI values were observed during ACLOUD due to a clearly separated cloud base from the surface and the absence of dense fog as observed during AFLUX. During N-ICE2015, the impact of less stable cloudy and cloud-free atmospheric states results in a more negative TNI distribution (Fig. 4b).

The difference in early summer between the cloudy modes observed during ACLOUD and N-ICE2015 relative to SHEBA (Fig. 4c) seems to be less related to atmospheric stability. When the surface temperature is fixed at zero degrees in the beginning of the melt season, further increasing atmospheric temperatures force a more stable near-surface layer causing a stronger temperature inversion, both supporting more neutral or even positive TNI as it seemed to be the case during SHEBA. However, as reported by Walden et al. (2017) (their Fig. 8) during N-ICE2015 and shown in Fig. 4b for ACLOUD, the ABL can also be unstable during spring and early summer, which yields rather neutral or slightly negative values of TNI.

## 4.2 Four-mode structure over different surface types

The ice floe camps during SHEBA and N-ICE2015 were operated on sea ice. In comparison, the low-level airborne observations during AFLUX and ACLOUD have been conducted over different (partly heterogeneous) surface types including compact sea ice, the fractional sea ice in the area of the MIZ, and open ocean. Thus, the airborne perspective reveals the effects of variable surface types on the downward, upward, and net thermal-infrared irradiances ($F_{\mathrm{tir}}^{\downarrow}$, $F_{\mathrm{tir}}^{\uparrow}$, and $F_{\mathrm{tir,net}}$) near the surface. In the following subsections we analyze the impact of surface type and clouds on the near-surface TNI (Subsection 4.2.1) and the downward and upward thermal-infrared irradiances (Subsection 4.2.2).

### 4.2.1 Near-surface thermal-infrared net irradiance (TNI)

An example of the impact of surface type on TNI is depicted in Figs. 5e and 5f, which show the PDFs of TNI counted within defined surface albedo bins. Here the albedo serves as an indicator of the different surface types. It is a key parameter determining the surface solar radiation budget. In this respect, Figs. 5e and 5f approximately relate the terrestrial (TNI) and the solar (albedo) surface radiative energy budgets to each other.

Not surprisingly, instead of two modes as observed over sea ice (Fig. 4), Figs. 5e and 5f reveal four main modes (red spots, indicated by numbers 1 to 4) of the near-surface TNI. These include the common two modes over sea ice (high surface albedo, 1 and 3), and in addition two further modes observed over open ocean (low surface albedo, 2 and 4). For AFLUX (Fig. 5e), the mode structure is more variable and noisy compared to a rather homogeneous TNI distribution for ACLOUD (Fig. 5f). For a given surface albedo, the variability of the near-surface TNI is larger for AFLUX than for ACLOUD, which is quantified by the median TNI values and their percentile ranges given in Table 2. This distinct difference between the twoi campaigns is caused by the less variable surface temperature and vertical temperature profile, and the mostly cloudy ABL with relatively invariant cloud base height and temperature, that prevailed during ACLOUD, compared to more variable and dynamic conditions during AFLUX. Furthermore, more data are available for ACLOUD (16 hours during low-level flights, while during AFLUX 6 hours were collected).

The surface albedo varies as a function of SZA, illumination and cloudiness, as well as surface (snow, ice, open ocean) properties. Furthermore, seasonal characteristics of surface and atmospheric properties impact the mode structure as discussed in Section 4.1. The footprints of these effects become apparent in Figs. 5e and 5f. For example, during AFLUX the median TNI values of the cloudy modes appear quite different over sea ice and open ocean, the difference between the modes ranges between $40\,\mathrm{W\,m^{-2}}$ and $50\,\mathrm{W\,m^{-2}}$. The same holds true for the cloud-free modes observed during AFLUX. However, during ACLOUD, these differences of the median values of the TNI modes observed over sea ice and open ocean are smaller (10–$20\,\mathrm{W\,m^{-2}}$). This behavior is partly caused by larger differences between the surface temperatures of sea ice and open ocean prevailing during AFLUX, compared to ACLOUD (Figs. 2c and 2d). In case of a large horizontal surface temperature gradient between sea ice and open ocean, the upward thermal-infrared irradiances emitted by the two surfaces differ significantly, which results in a larger spread of TNI mode values. Specifically, during AFLUX, horizontal surface temperature gradients between sea ice and open ocean of up to $25\,^{\circ}\mathrm{C}$ were observed, while during the second half of ACLOUD (melt period) the surface temperature differences between sea ice and open ocean were much reduced to values up to $6\,^{\circ}\mathrm{C}$.

However, the seasonal differences of horizontal surface temperature gradients between sea ice and open ocean cannot fully explain the observations. Using the Stefan-Boltzmann law for cloud-free conditions over open ocean and assuming typical surface and atmosphere temperatures, a typical TNI value around $-150\,\mathrm{W\,m^{-2}}$ is estimated for AFLUX, while mostly values of down to $-120\,\mathrm{W\,m^{-2}}$ were observed. The theoretically estimated strongly negative values of TNI values are not met by the observed ones because the thermodynamic adjustment of the atmosphere in the MIZ has to be taken into account as well. The cold air moving from the sea ice warms before reaching the open ocean, as shown in Fig. 3, compensating on average up

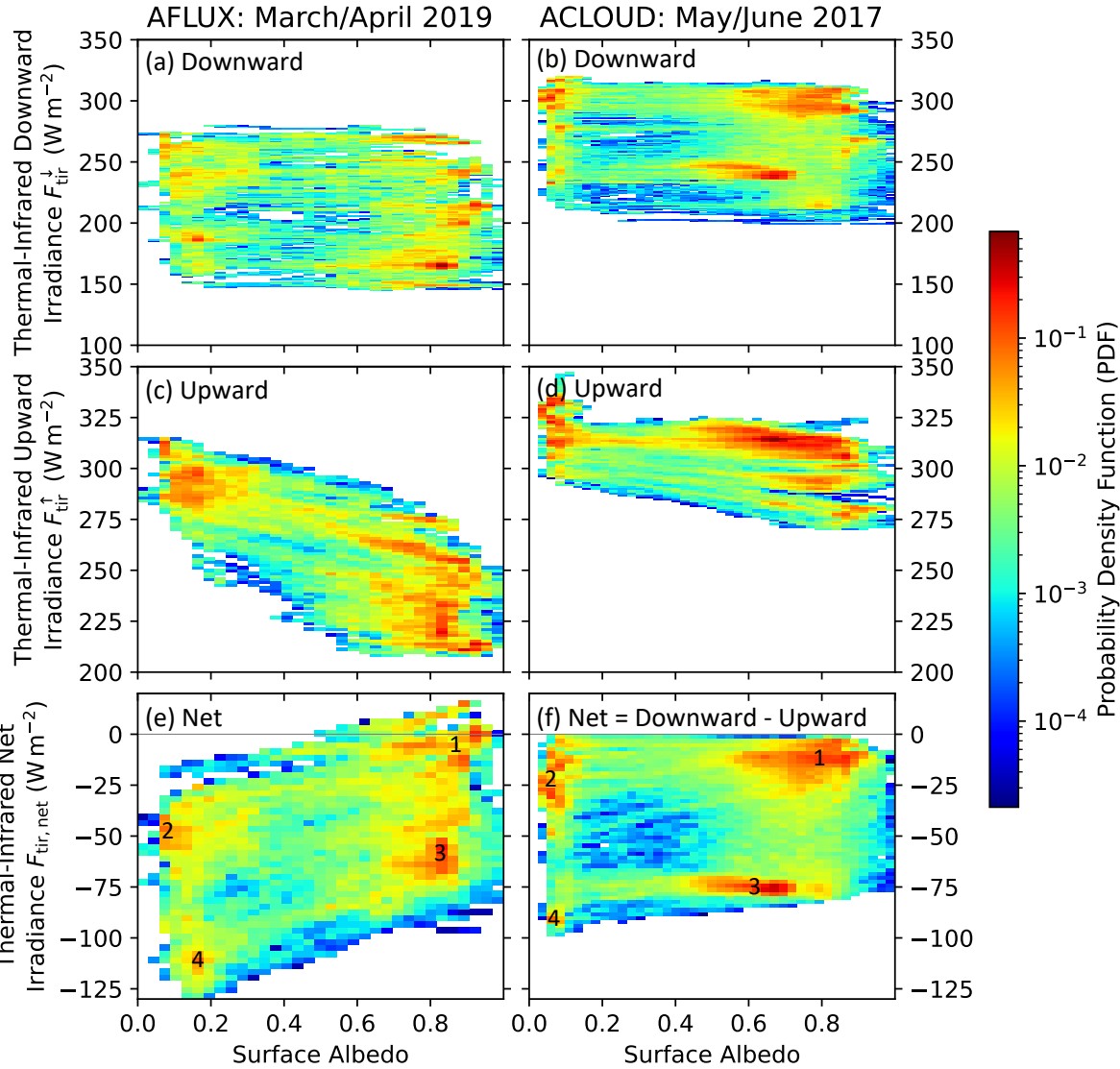

**Figure 5.** Two-dimensional PDFs of thermal-infrared downward (a) and (b), upward (c) and (d), and net (e) and (f) irradiances ($F_{\mathrm{tir}}^{\downarrow}$, $F_{\mathrm{tir}}^{\uparrow}$, and $F_{\mathrm{tir,net}}$) as a function of all-sky surface albedo, as observed during AFLUX (a), (c), and (e), and ACLOUD (b), (d), and (f). Panels (e) and (f) have been adapted from Wendisch et al. (2023). The numbers in (e) and (f) indicate cloudy modes (1 and 2) and the cloud-free modes (3 and 4) over sea ice (1 and 3) and open ocean (2 and 4), respectively.

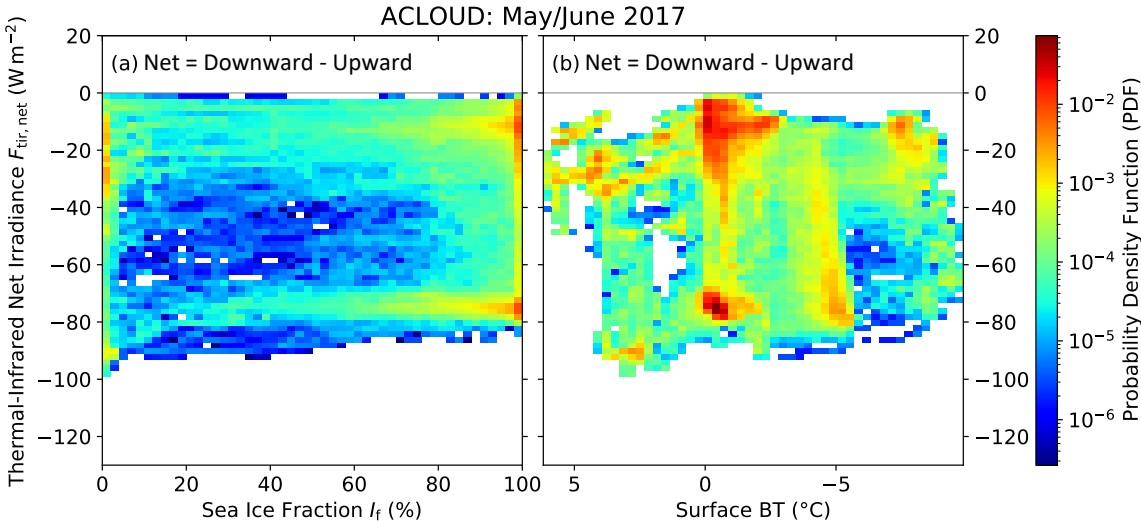

**Figure 6.** Two-dimensional PDFs of thermal-infrared net irradiances ($F_{\text{tir,net}}$) as a function of (a) sea ice fraction ($I_f$), and (b) surface brightness temperature (BT), observed during ACLOUD. The abscissa of panel (b) is reversed to highlight the position of the modes similar to Fig. 5.

to $30\,\mathrm{W\,m^{-2}}$ of TNI in cloud-free conditions during AFLUX. Similarly, warm air masses moving onto the sea ice will cool before reaching the cold sea ice surfaces.

Additionally, Fig. 5e shows that during ACLOUD, with less horizontal advection, the cloudy mode over open ocean splits into two sub-clusters, one representing the effects of the still cold air advection-dominated May (TNI around $-30\,\mathrm{W\,m^{-2}}$) and

5    a second one determined by the warm and neutral period of the campaign (around $-15\,\mathrm{W\,m^{-2}}$).

In Fig. 6, the sea ice fraction and the surface BT are used as independent variables on the abscissa to characterize the surface types, instead of surface albedo as in Figs. 5e and 5f. In particular, from a physical point of view, TNI has a closer link to the surface BT than to the surface albedo, which mostly considers the solar spectral range that is not directly relevant for the TNI as BT determines the upward component of the thermal-infrared radiation budget. In Fig. 6a, the modes are clearly clustered

10    around $I_f \approx 0\,\%$ (open ocean) and $I_f \approx 100\,\%$ (sea ice). Using the surface BT on the abscissa, links the surface emission to the TNI structure (Fig. 6b). In general, the four-mode structure becomes obvious. The values around $0\,^\circ\mathrm{C}$ belong to open ocean or melting sea ice, whereas the negative values of surface BT characterize the sea ice. Over sea ice (low values of BT) the cloudy mode is shifted to lower values of BT, compared to the cloud-free mode.

### 4.2.2 Near-surface downward and upward thermal-infrared irradiances

15    To discriminate the effects of surface albedo and clouds, we further consider Figs. 5a to 5d, which depict the pdfs of downward (a and b, cloud influence) and upward (c and d, surface albedo impact) thermal-infrared irradiances measured during the low-

level flights of AFLUX and ACLOUD. For both campaigns the cloudy and the cloud-free modes are imprinted in both the downward and upward thermal-infrared irradiances. Over sea ice this result is not surprising due to multiple scattering effects between the highly reflecting sea ice surface and the highly reflecting cloud base that mixes the two impact factors (surface albedo and clouds). Also over the open ocean, the four modes are obvious. To quantify the cloudy modes and their variability in the downward, upward, and net thermal-infrared irradiances ($F_{\text{tir}}^{\downarrow}$, $F_{\text{tir}}^{\uparrow}$, and $F_{\text{tir,net}}$), Table 2 gives the corresponding median and percentile values.

| | $F_{\text{tir}}^{\downarrow}$ [W m$^{-2}$] | | | $F_{\text{tir}}^{\uparrow}$ [W m$^{-2}$] | | | $F_{\text{tir,net}}$ [W m$^{-2}$] | | |
|---|---|---|---|---|---|---|---|---|---|
| | Median | 25pct | 75pct | Median | 25pct | 75pct | Median | 25pct | 75pct |
| AFLUX (open ocean) | 239 | 224 | 251 | 292 | 286 | 297 | -52 | -68 | -43 |
| AFLUX (sea ice) | 215 | 198 | 251 | 244 | 219 | 257 | -12 | -37 | -2 |
| ACLOUD (open ocean) | 302 | 290 | 308 | 323 | 313 | 331 | -24 | -31 | -15 |
| ACLOUD (sea ice) | 295 | 285 | 304 | 311 | 305 | 314 | -13 | -19 | -10 |

**Table 2.** Median, and percentile (pct, 25 % and 75%) values (in units of W m$^{-2}$) of the cloudy modes (defined as LWP > 5 g m$^{-2}$) of the downward, upward, and net thermal-infrared irradiances ($F_{\text{tir}}^{\downarrow}$, $F_{\text{tir}}^{\uparrow}$, and $F_{\text{tir,net}}$) measured during low-level flights during AFLUX and ACLOUD, as observed over different surface types (open ocean and sea ice) (see Fig. 5).

## 4.3 Effect of water vapor

Water vapor absorbs and emits thermal-infrared radiation and acts as one of the most important greenhouse gases. To interpret the role of water vapor in the development of the four-mode structure of the thermal-infrared radiative field close to the ground, we have quantified its influence on downward, upward, and net thermal infrared irradiances ($F_{\text{tir}}^{\downarrow}$, $F_{\text{tir}}^{\uparrow}$, and $F_{\text{tir,net}}$). For this purpose, we have performed radiative transfer simulations based on representative thermodynamic and cloud input values, over open ocean and sea ice. In the simulations we have switched on and off clouds and water vapor. The simulations yielded downward, upward, and net thermal-infrared irradiances for the wavelength range of 4–100 µm. The main input consisted of average thermodynamic profiles obtained from dropsondes launched from the aircraft during the campaigns (Fig. 3a and 3b). All simulations were performed for cloud-free and cloudy conditions. In the latter case, the assumed cloud was located at 400–600 m altitude with an LWP of 50 g m$^{-2}$ and a cloud droplet effective radius of 8 µm. These are typical values observed during the campaigns. The results of simulations are shown in Fig. 7. The downward thermal-infrared irradiance (Fig. 7a) is only slightly influenced by water vapor in the presence of clouds (gray bars), which dominate the below-cloud thermal-infrared radiative field. However, in cloud-free conditions (comparison of solid and hatched yellow bars), water vapor exerts a major influence (100 W m$^{-2}$ to 150 W m$^{-2}$) on downward thermal-infrared irradiances, which is not surprising considering the role of water vapor as a major greenhouse gas. Consequentially, the simulations also indicate that in an atmosphere with no water vapor, the effect of clouds (comparison of gray and yellow hatched bars) would be even larger at approximately

$200\,\mathrm{W\,m^{-2}}$ compared to the impact when water vapor is also present ( $75\,\mathrm{W\,m^{-2}}$ when comparing solid gray and yellow bars).For the average conditions assumed in the sensitivity simulations, the upward thermal-infrared irradiances at the surface are determined by the skin temperature, and therefore all considered scenarios yield the same values of $F_{\mathrm{tir}}^{\uparrow}$ (Fig. 7b). As a consequence, the net irradiances ($F_{\mathrm{tir,net}}$) are driven by the downward thermal-infrared irradiances ($F_{\mathrm{tir}}^{\downarrow}$) and the primary finding that the water vapor influence is negligible in the presence of clouds remains. Similarly, the net irradiances show that the influence is significant in cloud-free conditions.

## 5 Impact of synoptically driven meridional air mass transports on thermal-infrared radiation at the surface

The processes discussed above explain most of the variability of the TNI mode structure. They become particularly relevant, when air masses are transported from one temperature and/or surface regime into another, which typically happens during MCAOs and WAIs. Therefore, we investigate effects on surface TNIs that are caused during transformation processes that air masses experience during meridional transport in more detail in this section. WAIs (on-ice flow) into, and MCAOs (off-ice flow) out of the Arctic are typical examples of such meridional transports, both are driven by synoptic features. In a first part of this section, the method applied to identify MCAOs and WAIs is described. Furthermore, their effects on downward thermal-infrared irradiance at the surface are discussed. This is followed in the second and third parts of this section by a discussion of several case studies observed during AFLUX and ACLOUD, looking at the quantitative impact of WAIs and MCAOs on the surface TNI.

### 5.1 Influence on downward thermal-infrared irradiance at the surface in cloud-free conditions

To identify WAIs and MCAOs, the difference between the temperatures of the surface and the atmosphere at the geopotential height of $850\,\mathrm{hPa}$ is calculated from ERA5 reanalysis data (Hersbach et al., 2020) for the area of the Fram Strait west of Svalbard. The resulting parameter is commonly called the MCAO index (Papritz and Spengler, 2017; Knudsen et al., 2018). It should be noted that there is a general correlation between the MCAO index and the TNI, with increased TNI during WAI and decreased TNI during MCAO. Positive values of the MCAO index are characteristic for MCAOs, strongly negative MCAO values yield indications of WAIs. Therefore, the MCAO index can also be used to discriminate between mostly synoptically driven conditions (here MCAOs and WAIs), where thermodynamic adjustments become important, and locally determined situations when the ABL is in equilibrium.

The resulting time series of the MCAO index derived from the ERA5 reanalyses during the AFLUX and ACLOUD campaigns is depicted in Figs. 8a and 8b, respectively. During AFLUX (Fig. 8a), frequent switches between off-ice (positive MCAO index, blue-shaded) and on-ice (red-shaded) flow patterns occurred, caused by low-pressure systems moving from the south of Greenland towards Svalbard. For ACLOUD (Fig. 8b), Knudsen et al. (2018) used the MCAO index to discriminate between three synoptic periods: (i) a cold period containing two weak MCAO events during the last third of May 2017, (ii) a WAI lasting from the beginning to mid of June 2017, and (iii) a rather neutral period thereafter the end of June of 2017.

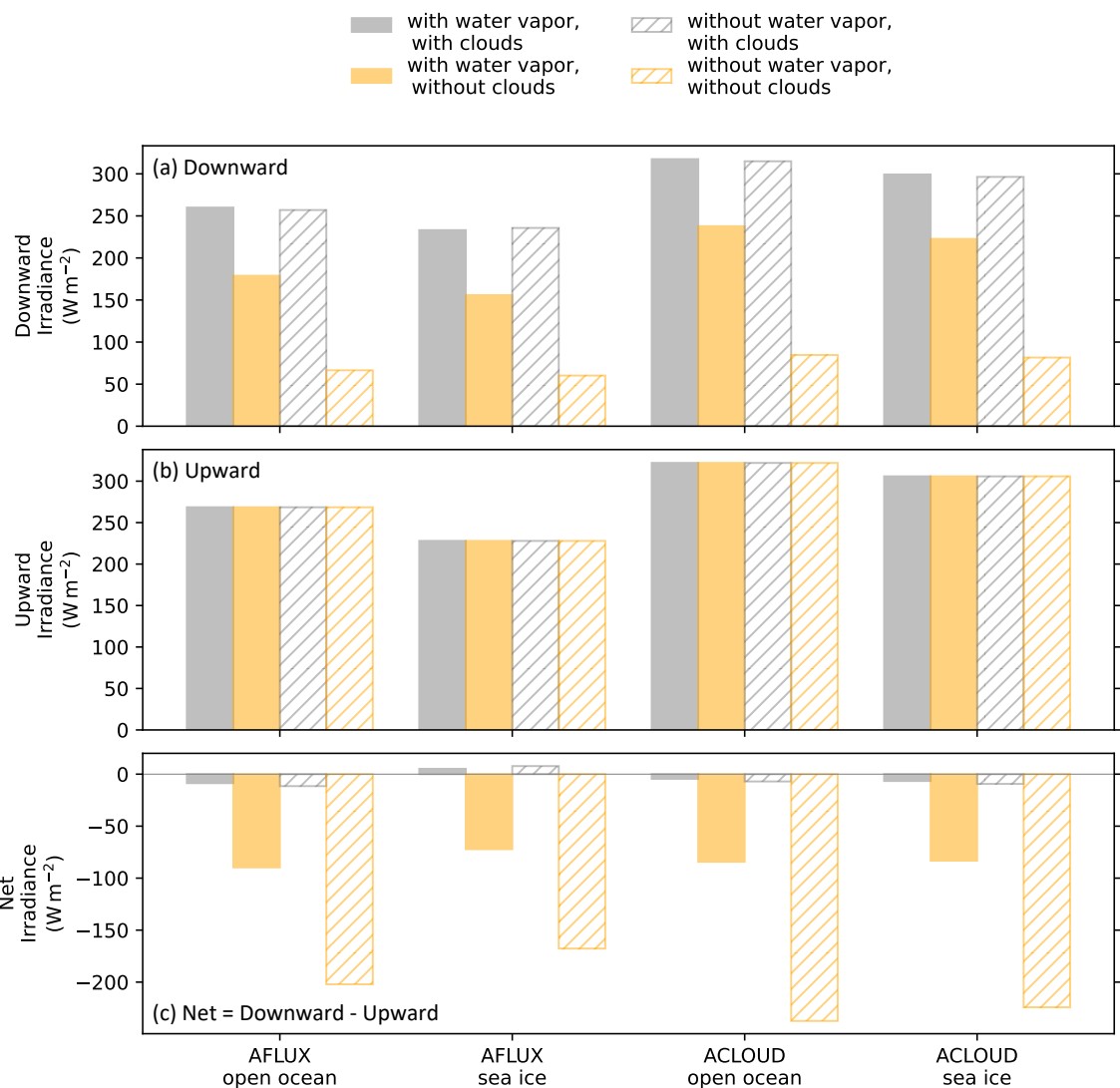

**Figure 7.** Simulated thermal-infrared downward (a), upward (b), and net (c) irradiances ($F_{\mathrm{tir}}^{\downarrow}$, $F_{\mathrm{tir}}^{\uparrow}$, and $F_{\mathrm{tir,net}}$) at the surface for different scenarios with and without water vapor and clouds.

To investigate the influence of these synoptically driven phenomena (MCAOs, WAIs) on the near-surface TNI, in a first step the time series of the downward thermal-infrared irradiance at the surface in cloud-free conditions, $F_{\mathrm{tir,cf}}^{\downarrow}$, was simulated (Figs. 8c and 8d). The radiative transfer calculations were performed using data from the permanent ground station in Ny-Ålesund (AWIPEV research base), which provided daily radiosonde data, and measurements from dropsondes released from the aircraft. For comparison, a climatology of $F_{\mathrm{tir,cf}}^{\downarrow}$ was extracted from ERA5, in the location of the aircraft flights. At

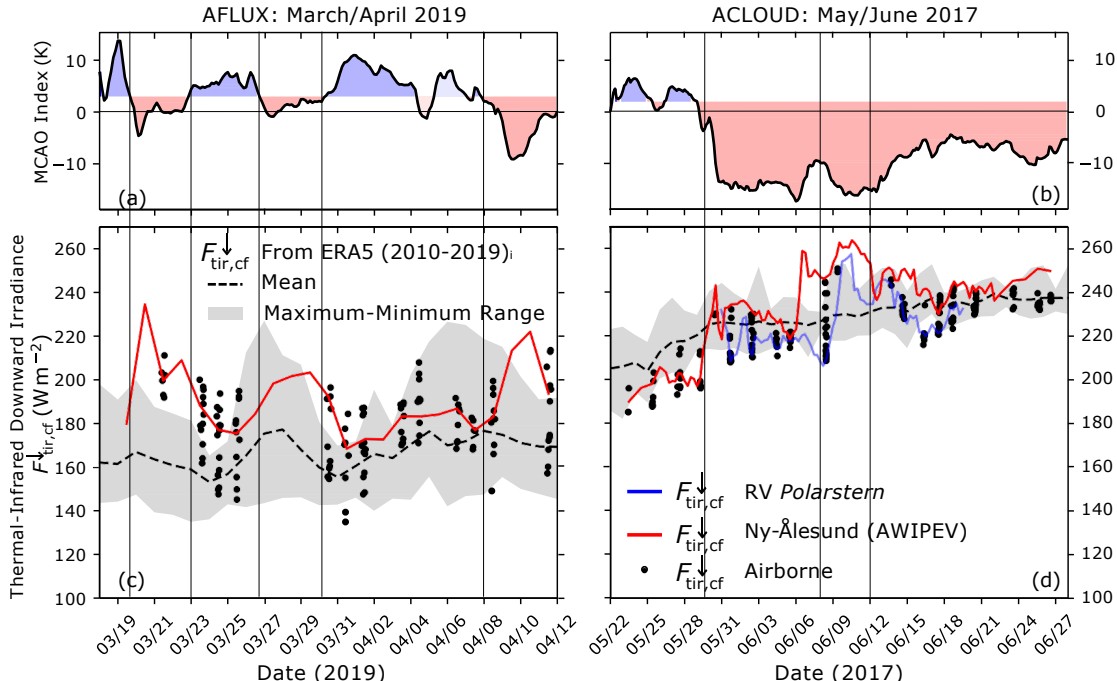

**Figure 8.** (a) and (b) Time series of the marine cold air outbreak (MCAO) index calculated on the basis of thermodynamic profiles from ERA5 reanalysis data in the area between $77°$ N and $80°$ N and $2.5°$ E to $10°$ E (cloud-free conditions, black line), and using a threshold of $3\,°C$ to characterize MCAO-dominated scenes (blue shading). (c) and (d) Time series of simulated downward thermal-infrared downward irradiances in cloud-free conditions at the surface, $F_{\mathrm{tir,cf}}^{\downarrow}$. Profile data of thermodynamic parameters from four sources were used as input for the radiative transfer simulations: ERA5-based ten-year climatological mean (2010-2019) values (black dashed lines and gray shading), data from radiosondes launched at RV *Polarstern* (blue line) and Ny-Ålesund (AWIPEV research base, red line), and airborne *in situ* profile measurements collected during the aircraft campaigns (dropsondes) AFLUX and ACLOUD (full black dots).

the beginning of AFLUX (20 to 23 March 2019, weak WAI), the Ny-Ålesund (red line) and dropsonde (full black dots) measurement-based simulations indicate a roughly $40\,\mathrm{W\,m^{-2}}$ higher $F_{\mathrm{tir,cf}}^{\downarrow}$ at the surface compared to the respective data from the ERA5-based climatology (Fig. 8c). With the beginning and intensification of a MCAO between 23 and 26 March 2019, the atmospheric and surface temperatures steadily decreased, and so did the values of $F_{\mathrm{tir,cf}}^{\downarrow}$. The simulated values of $F_{\mathrm{tir,cf}}^{\downarrow}$ based
5  on dropsondes released from aircraft include observations over a wider range of conditions (sea ice and open ocean, warm and cold air masses) (full black dots) and therefore, vary by up to $50\,\mathrm{W\,m^{-2}}$. The respective variability is mostly covered by the maximum range of the ERA5 climatology (gray shaded). The same holds for the second main MCAO period during AFLUX (30 March to 8 April 2019). In the final WAI period during AFLUX (8 to 12 April 2019), the measurement-based values of $F_{\mathrm{tir,cf}}^{\downarrow}$ increased over the climatological mean values.
10    At the beginning of ACLOUD (until 29 May 2017), during two weak MCAO situations, the measurement-based (Ny-Ålesund, airborne), simulated values of $F_{\mathrm{tir,cf}}^{\downarrow}$ are smaller by approximately $20\,\mathrm{W\,m^{-2}}$ than those derived from the ERA5

climatology (Fig. 8d). Due to weaker vertical temperature gradients (Fig. 3b), the range of changes of $F_{\mathrm{tir,cf}}^{\downarrow}$ decreased significantly compared to AFLUX. In the subsequent WAI period (29 May to 7 June 2017), the simulated values of $F_{\mathrm{tir,cf}}^{\downarrow}$ increased by up to $30\,\mathrm{W\,m^{-2}}$, mainly due to temperature advection in the free troposphere, while a second influx of warm and moist air between 8 and 12 June 2017 increased $F_{\mathrm{tir,cf}}^{\downarrow}$ even further, highlighting the importance of increased moisture in the advected
air mass.

     In general, the results from both campaigns clearly indicate the significant impact and delicate interplay of synoptically forced WAIs or MCAOs on $F_{\mathrm{tir,cf}}^{\downarrow}$. In particular, the conditions during AFLUX appeared to be driven mostly by synoptic processes whereas ACLOUD was less influenced by the synoptically driven meridional air mass transports due to weaker temperature contrasts between the open ocean and sea ice. Nevertheless, during both campaigns the daily variability in cloud
distributions and properties in the area clearly mask trends in the TNI linked to synoptic processes. The thermodynamic atmospheric background is of decisive importance (Stapf et al., 2021a), which underlines that both aspects (synoptic scale and local effects) have to be analyzed separately.

     The differences between the simulated values of downward irradiances based on radiosonde launches from RV *Polarstern* (in the sea ice) and Ny-Ålesund (blue and red lines), and those released from aircraft emphasize the regional variability of atmo-
spheric and surface conditions during WAIs and MCAOs. The regional variability of the downward irradiances is also obvious from the aircraft measurements alone. For example, for 25 March downward irradiance values range from about $130\,\mathrm{W\,m^{-2}}$ to $190\,\mathrm{W\,m^{-2}}$ (or on 12 March from $150\,\mathrm{W\,m^{-2}}$ to $215\,\mathrm{W\,m^{-2}}$). This variability is linked to air mass transformations, which can not be observed at single ground-based stations instantaneously. Therefore, airborne observations along the trajectory of the transported air masses are analyzed in the following two subsections to study the coupling of these transition processes and
the TNI.

## 5.2   Influence of warm air intrusions: On-ice flow

In general, air mass transformation during WAIs tends to cool the initially warm air mass, leading to condensation, enhanced radiative cooling, and corresponding vertical mixing, while also warming the surface via TNI, which eventually pushes the air mass towards thermal equilibrium with the surface. During AFLUX and ACLOUD, three distinct events of WAIs with
respective on-ice flows were sampled and analyzed in detail to observe these processes: 2 June 2017, 8 April 2019, and 21 March 2019. In Figures 9a to 9c, the corresponding Moderate Resolution Imaging Spectroradiometer (MODIS) satellite images including the TNI measured below clouds (colored dots) are presented for these three cases. Figures 9d to 9f depict corresponding temperature profiles during the flights derived from dropsondes. During the airborne observations conducted on 2 June 2017 (ACLOUD), a strong south-westerly on-ice flow caused by a high pressure system located southeast of Svalbard
transported optically thick clouds northward over the MIZ (Fig. 9a). This case has been described in detail by Chechin et al. (2023) with a focus on turbulent cloud and surface related processes. During AFLUX, on 8 April 2019, a second case of an on-ice flow characterized by weak winds from the south and fog in the first few kilometers north of the ice edge was probed (Fig. 9b). On 21 March 2019 a south-easterly advection of thick mid-level clouds due to a low-pressure system to the east of

Greenland was observed (Fig. 9c). The three cases differ in the way the thermodynamic profiles transform along the on-ice flow (Figs. 9d-f). Furthermore, the altitude of the transported clouds was different for the three cases.

### 5.2.1    2 June 2017: Weak surface temperature contrast, thin low-level clouds

On 2 June 2017 (Fig. 9a), a weak surface temperature gradient of 2-3 °C between sea ice floes in the MIZ and the open ocean south of the MIZ prevailed (Fig. 9d). Over the ocean, the temperature inversion at cloud top was observed at around 750 m altitude (Fig. 9d, brown, red, yellow, light green lines, labeled 1-4). However, further North over the MIZ and the sea ice, the cloud top temperature inversion was observed at lower altitudes between 250 m and 400 m (lines of dark green, blue, purple, black color in Fig. 9d, labeled 5-8). The lowering of the cloud top inversion appeared rather abruptly over the sea ice edge, while the cloud top temperature remained almost constant during the transport from the open ocean towards the MIZ. Because the cloud base temperature was consistently colder than the near-surface air temperature, the observed TNI measured during the low-level sections appeared mostly negative (Fig. 9a), in particular over the open ocean. However, due to the permanent presence of clouds, the surface radiative cooling was moderate with values between $-5\,\mathrm{W\,m^{-2}}$ and $-15\,\mathrm{W\,m^{-2}}$. Thus, the surface lost radiative energy in the thermal-infrared wavelength range despite the advection of warm air. This energy loss may contribute to the ongoing transformation of the warm air mass, although in this particular case since there is still a surface loss of radiative energy, then to first order, the cloud base is being warmed by the surface, which inhibits the cloud development and air mass transformation. Most of the transformation is likely caused by the cloud top radiative cooling, and its subsequent mixing due to buoyancy considerations. In this case the temperature profiles also suggest that the cloud is thermodynamically coupled to the surface, with vertical mixing linking the cloud and surface layers as described in more detail by Chechin et al. (2023). These authors considered this case with a focus on the turbulent boundary layer structure pointing to the interplay between surface processes and slight cloud generated turbulence.

### 5.2.2    8 April 2019: Strong surface temperature gradient, fog

During the weak on-ice flow prevailing on 8 April 2019 (Fig. 9b), a strong surface temperature gradient between the open ocean and sea ice of more than 10 °C was observed (Fig. 9e). With a light flow from the south, the warm and moist air in the lowest 800 m cooled and mixed with the cold near-surface air over the MIZ causing condensation and formation of a low cloud layer. Further to the north, under stronger near surface stratification, this layer was more suppressed vertically, comprising a shallow fog less than 200 m deep. At the top of this cloud/fog, strong radiative cooling induced the development of a temperature inversion (Fig. 9e), which further enhanced the condensation. The dense, low-level cloud caused near-neutral TNI values close to the surface over the MIZ (labeled 5 and 6) due to the near temperature equilibrium between effective cloud base and surface. However, the cloud layer was still decoupled from the surface by a very shallow surface-based inversion at this time. Radiative transfer simulations using the profile information right near the surface confirm that the clouds were very weakly warming the surface at this time, with small positive TNI, which helped to further diminish the near-surface stratification that decoupled the cloud from the surface. Further to the north, the temperature gradient between the warm atmosphere aloft and the cold surface was nearly 5 °C, which confined the condensation to a thinning layer of fog right near the surface. This diminishing

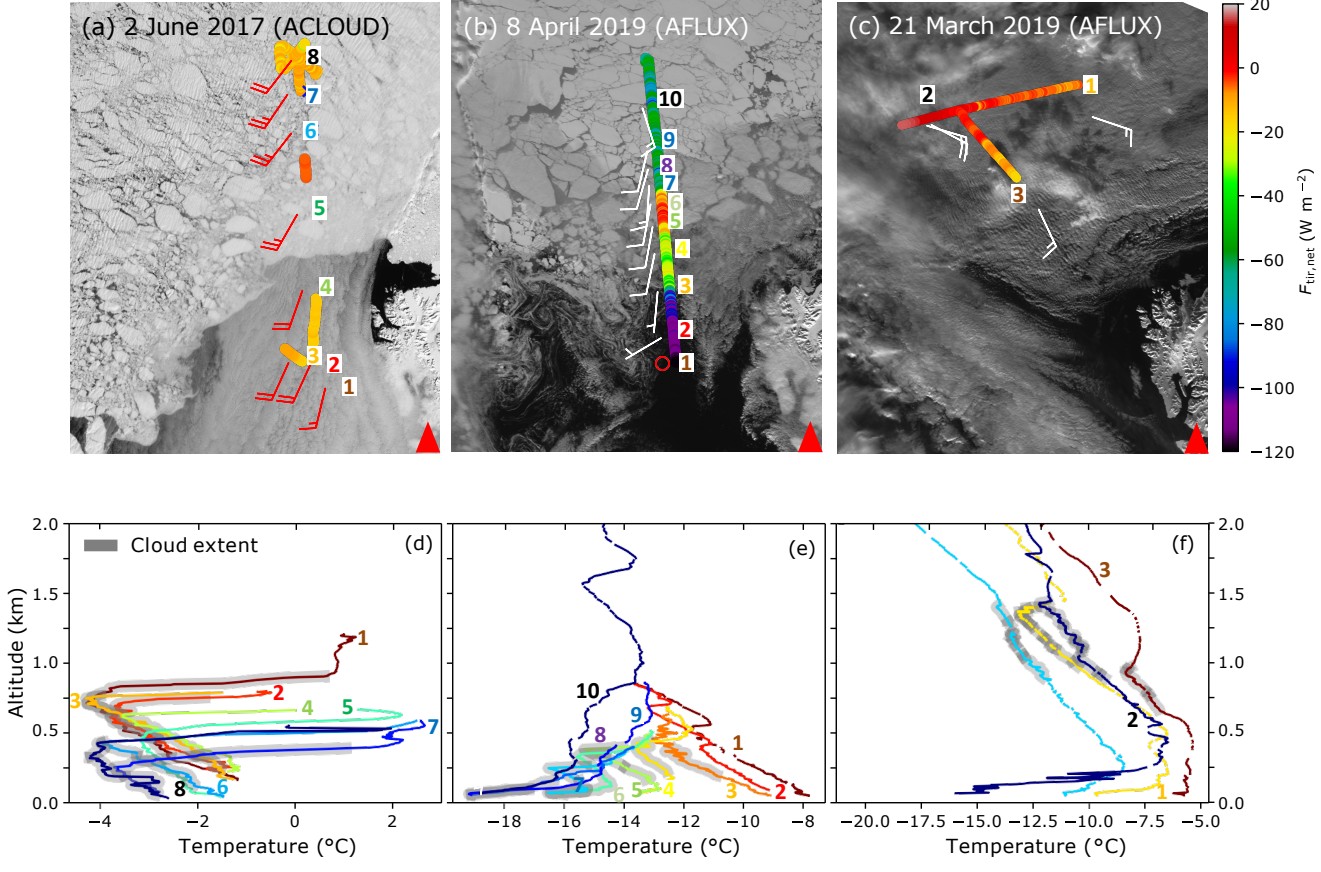

**Figure 9.** MODIS satellite images (MODIS Characterization Support Team, 2017) of three cases of on-ice flow observed on (a) 2 June 2017, (b) 8 April 2019, and (c) 21 March 2019. The overlaid colored open dots indicate the values of the measured TNI along the flight path during low-level flight sections. The colored numbers mark the locations where *in situ* profile observations of wind speed and direction (at cloud height, feeders) and temperature were taken. These labels were slightly shifted away from the flight track (open circles). The colored number in (a-c) are used in (d-f) to mark the temperature profiles. Red full triangles indicate the location of Ny-Ålesund at Svalbard. (d-f) *in situ* temperature profiles color-coded by the locations given by the number labels in panels (a-c). The cloud extent is flagged with a gray shading.

fog also had a diminished impact on the TNI, which approached values of –60 W m$^{-2}$ to the north of the MIZ that are typical of cloud-free conditions under a surface-based temperature inversion.

### 5.2.3 21 March 2019: Strong surface temperature contrast, thick elevated clouds

In the first two WAI cases discussed above, no strong thermal-infrared radiative warming of the surface (positive TNI) was detected, especially over the sea ice. However, on 21 March 2019 the in-flowing warm and moist air actually warmed the surface radiatively (Fig. 9c). In this case, optically thick clouds at altitudes between 900 m and 1400 m were embedded in a warm air mass remaining from a strong WAI that took place the day before (Fig. 9f). During the low-level flights, the surface temperatures ranged between –6 °C and –15 °C. The cloud base temperatures varied between –7 °C to –10 °C. Consequently, the surface TNI varied over a range of almost 25 W m$^{-2}$, peaking at 17 W m$^{-2}$, which represents a strong thermal-infrared warming effect of the surface (gain of radiative energy by the surface). In the time series of surface BT (not shown), this event exhibits the warmest sea ice temperatures observed during AFLUX. After about an hour, the light-blue temperature profile was observed in the measurement area (Fig. 9f). The atmospheric temperature decreased above the near-surface inversion by up to 3 °C within one hour, illustrating the shifting conditions during that day. Consequently, for the last low-level section towards the southeast a neutral TNI was observed, steadily decreasing as the aircraft approached the ocean with warmer surface temperatures relative to the cloud temperatures. Despite the weak solar insolation on this day, strongly positive total (solar plus thermal-infrared) net irradiances (not shown) were observed towards the west, due to a decreasing cloud optical thickness enhancing the solar transmissivity.

### 5.2.4 Surface cooling or warming during WAIs

The analysis of the three cases highlights the importance of air mass transformation processes in controlling the amount of radiative energy that is absorbed by the sea ice and the open ocean in the MIZ. These processes shape the thermodynamic profiles in combination with the cloud physical and dynamical properties, and sometimes the surface temperature. Lifted clouds embedded in warmer air masses over colder sea ice, which are disconnected from the ground by a surface-based temperature inversion (21 March 2019), have the potential to warm the surface more strongly than thin boundary layer clouds (2 June 2017) or near-surface fog (8 April 2019). This is mainly due to the higher cloud base temperature of the lifted clouds. In general, a surface-based temperature inversion may contribute to a radiatively driven surface warming.

### 5.3 Impact of marine cold air outbreaks during AFLUX: Off-ice flow

During AFLUX, a series of MCAOs (Fig. 8a) transported cold and dry air masses from the inner Arctic by a northerly flow from the sea ice towards the open ocean, where high surface fluxes of heat and moisture prevailed within the unstable ABL, causing strong convection and roll cloud formation (Brümmer, 1997; Pithan et al., 2018). Three consecutive MCAOs were observed between 23 and 27 March 2019, which were steered by a low-pressure system located east of Svalbard. Trajectory analyses from these cases (not shown) indicate that the air masses originated from further south, namely from the Norwegian and Barents Seas, advected around the low-pressure system. The trajectories likely explain the mid-level, optically thick clouds over sea ice observed on most of 24 and 25 March and intermittently on 25 March. During a second series of MCAOs, starting

at the end of March 2019 (Fig. 8a), the air mass was transported from the inner Arctic without forming low-level clouds, instead with thin cirrus present.

As an example, in Figure 10a, the satellite image of the MCAO observed on 25 March 2019 is documented. The TNI measured during the low-level flight sections along the flight track is included in this figure (colored dots). Due to the absence of local moisture sources over the cold, closed sea ice and the typical surface-based temperature inversion, cloud-free conditions would commonly be expected over the sea ice during MCAOs. However, because of the advected mid-level clouds (homogeneous cloud field west of the flight track), and the presence of leads in the windward direction steering the formation of low-level thin ABL clouds, the synoptic conditions were rather cloudy above sea ice on this particular day.

Figures 10b to 10d compare the near-surface downward (Fig. 10b), upward (Fig. 10c), and net (Fig. 10d) thermal-infrared irradiances measured along the low-level flight sections during three investigated MCAO cases during AFLUX (24, 25, 31 March 2019) as a function of geographic latitude. Above sea ice and the MIZ (geographical latitude larger than about 80°), the observed TNIs fluctuate between values typical for cloud-free conditions (–30 W m$^{-2}$ and –80 W m$^{-2}$) as observed on 25 and 31 March, and values between 0 W m$^{-2}$ and –10 W m$^{-2}$ measured below optically thick clouds on 24 March 2019. These values represent the two TNI modes over sea ice discussed in Fig. 5e. Closer to the sea ice edge, when the air masses approach the warm ocean surface of the MIZ, strong fluctuations with periodic extreme values of $F_{\mathrm{tir,net}}$ down to –120 W m$^{-2}$ (31 March 2019) are observed as a result of the variable surface temperature in the mixture of sea ice (cold) and open ocean (warm). Further downstream of the MIZ, the large negative TNIs decrease in magnitude towards the south as cloudiness increases and the temperature difference between the atmosphere (cloud base) and the surface diminishes. In the presence of evolving roll clouds, still strongly negative TNI values between –40 W m$^{-2}$ and –70 W m$^{-2}$ are observed over the ocean. Therefore, in addition to the strong upward turbulent fluxes over the warm open ocean, radiative energy fluxes further enhance the surface cooling of the ocean.

To investigate the transformation of the air mass during such transport events, the thermal-infrared downward irradiances $F_{\mathrm{tir,cf}}^{\downarrow}$ were simulated assuming cloud-free conditions, as a rough proxy for the temperature of the air mass (not shown). For all three cases, a generally increasing tendency of $F_{\mathrm{tir,cf}}^{\downarrow}$ from North to South towards the ice edge was simulated. This illustrates the warming of the cold, inner-Arctic air mass on its way southward. In reality, cloud-free situations are almost never observed in CAOs. On the contrary, usually dense roll clouds develop over open ocean and, as shown in Fig. 10a, they lead to significant increases in $F_{\mathrm{tir}}^{\downarrow}$. Both effects, warming of the atmosphere and increasing cloud fraction towards the South, surprisingly, compensate for the strong change of surface temperature (upward irradiance) resulting in a similar surface TNI over the partly cloud-free (optically thin clouds) closed sea ice and the cloudy open ocean. Altogether, the thermodynamic profile of the air mass adapts relatively slowly to the warmer ocean surface. Thereby, the TNI values increase continuously (Fig. 10d) with increasing distance from the ice edge, which is primarily caused by the slow adjustment of the ABL and cloud base temperature to the warmer surface temperature.

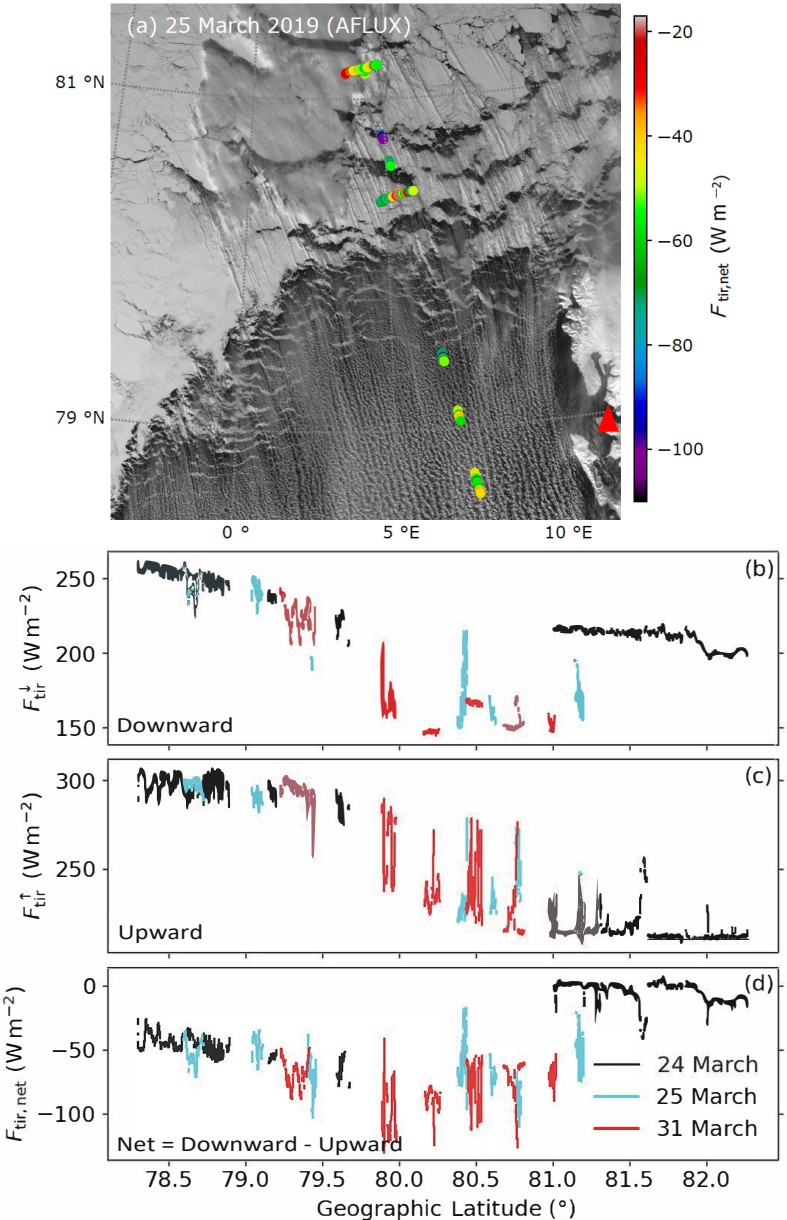

**Figure 10.** (a) MODIS satellite image (MODIS Characterization Support Team, 2017) of a MCAO on 25 March 2019, overlaid by the TNI measured during low-level flight sections. Observed (b) downward, (c) upward, and (d) net thermal-infrared irradiances ($F_{\text{tir}}^{\downarrow}$, $F_{\text{tir}}^{\uparrow}$, and $F_{\text{tir,net}}$) measured during the MCAOs on 24 (black line), 25 (blue line), and 31 March (red line) in 2019 as a function of latitude. The latitude of satellite-derived 80 % sea ice fraction isoline (Spreen et al., 2008) crossing the flight leg was observed between 79.6 ° N and 79.8 ° N. Sea ice was observed north, open ocean south of the 80 % sea ice fraction isoline. The red full triangles indicates the location of Ny-Ålesund at Svalbard.

# 6 Summary and conclusions

Low-level airborne observations of the near-surface thermal-infrared net irradiance (TNI) are discussed. The data have been collected over the heterogeneous environment of the MIZ, homogeneous sea ice, and open ocean northwest of Svalbard during the AFLUX and ACLOUD campaigns in spring and early summer. The TNI is driven by the sea ice concentration, the cloud properties, and the atmospheric thermodynamic transition between the warm ocean and the cold sea ice. Small-scale inhomogeneities of surface properties and thermodynamic profiles may cause complex interactions including radiative processes.

Above homogeneous sea ice, the common two-mode structure of the near-surface TNI field is subject to a seasonal shift toward more negative net irradiances and a higher frequency of occurrence of cloudy states compared to cloud-free situations from winter towards summer. The shift is caused by the non-linearity of the Planck emission (Stefan-Boltzmann law). Furthermore, the TNI field is linked to seasonal thermodynamic profile characteristics, which control the distribution of cloud-free and cloudy modes of net irradiance. For the area over and close to the MIZ, including heterogeneous surface conditions over the MIZ itself and the neighboring areas with homogeneous sea ice and open ocean surfaces, four separated modes of near-surface TNI are observed. On small horizontal scales in the MIZ, leads may cause large values of upward emitted irradiance and further special effects caused by the obvious change of surface temperature compared to the surrounding sea ice. On larger horizontal scales, the adjustment of the atmospheric thermodynamic conditions and cloud properties relocate the thermal-infrared TNI modes over the open ocean. Strongly negative TNI in cloudy as well as cloud-free conditions represent a special feature in the MIZ caused by the large difference between air/atmospheric and surface temperatures, especially during MCAOs.

The impact of warm air intrusions on the TNI in the MIZ is sensitive to the thermodynamic profiles along the flow and the vertical location of clouds, both of which are related to the transformation of air masses. We observed that the strongest direct warming potential of clouds in the thermal-infrared wavelength range is found for clouds included in an advecting warm air mass aloft, rather than fog embedded directly in strong surface-based temperature inversions below the advecting air mass. When MCAOs move over the open ocean, the TNI is shown to be surprisingly stable, although the individual effects of the changing surface and cloud conditions are large. In total, both effects, warming of the atmosphere and increasing cloud fraction almost compensate each other. The presence of leads/nilas upstream of the ice edge induces enhanced downward thermal-infrared irradiances via cloud initiation and ABL warming/moistening during MCAOs and, thus, likely contributes to a warming of the ice floes in the MIZ.

The MIZ, in addition to being complex, is actually a very important area. Radiative transfer and thermodynamic processes that play out in the complex environment of the MIZ are crucial for controlling the movement of the sea ice edge, and thus have important influence on sea ice areal extent. Aside from representing only a small area of the Arctic, observing and modeling the processes in this area is a challenge, but also an opportunity to improve the understanding of the TNI in the inner Arctic. The extreme changes of ABL thermodynamics and their impact on the TNI in the MIZ, likely overemphasize the importance of the thermodynamics. However, even slight shifts in the TNI modes above homogeneous sea ice underline the importance of the linkage between low-level atmospheric profiles and stability, in cloudy as well as cloud-free conditions.

It is known, that for climate models, reproducing the right regime of cloud and surface properties is a challenge. In light of the findings of this study, which show how particularly sensitive the surface TNI is to cloud and atmosphere properties, and considering insufficient statistics on real cloud properties over the entire inner Arctic, as well as diverging/variable estimates from satellites, models, and observations (Zygmuntowska et al., 2012; Cesana et al., 2012; Pithan et al., 2014; Achtert et al., 2020), conclusions on potential future cloud impacts and feedbacks in the Arctic are unclear. One key step towards bringing more clarity to the topic is to achieve greater consistency among the different observational and modeling perspectives that are brought to bear. The detailed information provided by the aircraft observations presented here can serve as an important tool to evaluate both how satellite products represent the spatial distribution of TNI across the sea ice edge, as well as how models reproduce the covariability and spatial interactions among clouds, the surface, and transforming Arctic air masses.

*Data availability.* The ACLOUD and AFLUX broadband irradiance and surface brightness temperature data are available from Stapf et al. (2019) and Stapf et al. (2021b), respectively. Broadband radiation and radiosonde data from the N-ICE2015 campaign have been taken from Hudson et al. (2016, 2017), for SHEBA from Persson (2011) and Moritz (2017). The ERA5 reanalyses were obtained from Hersbach et al. (2018a, b). Air temperature, relative humidity, and pressure *in situ* profile measurements from Polar 5 and Polar 6 aircraft can be downloaded from Hartmann et al. (2019), dropsonde data from Ehrlich et al. (2019a) and Becker et al. (2020). The radiosonde data measured at Ny-Ålesund and from RV *Polarstern* are from Maturilli (2020) and Schmithüsen (2017), respectively.

*Author contributions.* All authors contributed to the writing and editing of the article, and to the analysis and discussion of the results. JS and SB processed and merged the data, performed the radiative transfer simulations, and drafted the article, jointly with MW. MW, AE, and CL designed the observational basis of this study.

*Competing interests.* The authors declare that they have no conflict of interest.

*Acknowledgements.* We gratefully acknowledge the funding by the Deutsche Forschungsgemeinschaft (DFG, German Research Foundation) – Project Number 268020496 – TRR 172, within the Transregional Collaborative Research Center "ArctiC Amplification: Climate Relevant Atmospheric and SurfaCe Processes, and Feedback Mechanisms $(\mathcal{AC})^3$". The authors are grateful to AWI for providing and operating the two aircraft during ACLOUD and AFLUX. We thank the crews and the technicians of the Polar 5 and Polar 6 aircraft for excellent technical and logistical support. The generous funding of the flight hours for ACLOUD by AWI is greatly appreciated. Data provision by NCAR/EOL under the sponsorship of the National Science Foundation (https://data.eol.ucar.edu/) is acknowledged. The authors acknowledge Stephen Hudson and Lana Cohen of the Norwegian Polar Institute and Von P. Walden of Washington State University for supporting in handling the N-ICE2015 data. MDS was supported by a Mercator Fellowship with $(\mathcal{AC})^3$.

**APPENDIX: List of acronyms (in alphabetic order).**

| | |
|---|---|
| ABL | Atmospheric Boundary Layer |
| $(\mathcal{AC})^3$ | Arctic Amplification: Climate Relevant Atmospheric and Surface Processes, and Feedback Mechanisms |
| ACLOUD | Arctic CLoud Observations Using airborne measurements during polar Day |
| AFLUX | Airborne measurements of radiative and turbulent FLUXes in the cloudy atmospheric boundary layer |
| AWI | Alfred Wegener Institut, Helmholtz Zentrum für Polar- und Meeresforschung |
| AWIPEV | Alfred Wegener Institute for Polar and Marine Research (AWI) and the French Polar Institute Paul Emile Victor (IPEV) |
| BT | Brightness Temperature |
| LWP | Liquid Water Path |
| MCAO | Marine Cold Air Outbreaks |
| MIZ | Marginal Sea Ice Zone |
| MODIS | Moderate Resolution Imaging Spectroradiometer |
| N-ICE2015 | Norwegian Young Sea Ice Experiment |
| PASCAL | Physical feedback of Arctic ABL, Sea ice, Cloud and Aerosol |
| pct | Percentile |
| PDF | Probability Density Function |
| RV | Research Vessel |
| SHEBA | Surface Heat Budget of the Arctic Ocean |
| SZA | Solar Zenith Angle |
| TNI | Thermal-Infrared Net Irradiance |
| WAI | Warm Air Intrusion |

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
