# Peer review of "Effects of variable, ice-ocean surface properties and air mass transformation on the Arctic radiative energy budget"

_Atmospheric Chemistry and Physics, 2022_

## Author Comment (AC1)

**Response to Reviewer #1**

Thank you for your comprehensive and valuable comments and suggestions, which we have carefully considered. We very much appreciate your thoughts and time; your ideas really improved and enriched the manuscript! Please look at our responses and the revised manuscript.

The text of your review is copied below in *Calibri black font italic style*; replies are given in black style.

*Reviewer #1: The paper describes the TSI collected during two campaigns ACLOUD and AFLUX (and two additional experiments) over a sea ice/open ocean transition area in the Arctic, and attempts to link the different infrared radiative regimes with air and surface temperature, air mass advection, sea ice coverage. While I should recognize the outstanding value of the dataset, I believe the water vapor content contribution is not highlighted sufficiently, as it is one of the main driver of the effective emissivity of the atmosphere, which in turn determines the downelling component of the LW radiation. Some specific comment on this are given in the detailed review I reported below.*

**Reply:** Thank you for emphasizing the need to clarify the role of water vapor in our investigations. To consider your request, we have included a new subsection (4.3 Effect of water vapor) and a new Figure 7 in the revised manuscript. The new subsection 4.3 aims at quantifying the impact of water vapor on the downward, upward, and net thermal-infrared irradiances close to the surface. For this purpose, we have performed radiative transfer simulations based on representative thermodynamic input and cloud properties, over open ocean and sea ice. In the simulations we have switched on and off clouds and water vapor.

The radiative transfer simulations yielded downward, upward, and net irradiances ($F_{\text{tir}}^{\downarrow}$, $F_{\text{tir}}^{\uparrow}$, $F_{\text{tir,net}}$) for the wavelength range of 4–100 μm. The main input consisted of average thermodynamic profiles obtained from dropsondes launched from the aircraft during the campaigns (e.g., temperature profiles, see Figs. 3a and 3b in the manuscript). These profiles are separated for campaign and surface type (sea ice and open ocean areas, defined as regions with a sea ice concentration above 90 % and below 10 %, respectively, derived from daily sea ice concentrations maps from Spreen et al., 2008), and are extended to altitudes above flight level by corresponding average radiosonde profiles obtained over Ny-Ålesund.

All simulations are performed for cloud-free and cloudy conditions. In the latter case, the assumed cloud is located at 400–600 m altitude with a liquid water path (LWP) of 50 g m$^{-2}$ and a cloud droplet effective radius of 8 μm. These are reasonable assumptions that have proven representative during the campaigns. The results of the simulations are shown below in Figure 1 of this reply, which is included as the new Figure 7 in the revised manuscript. We will come to the interpretation of Figure 7 later in this reply.

*Reviewer #1: The author proven their in depth knowledge of the mechanisms driving the TSI, while some additional efforts trying to isolate individual/relative contributions of them, might be spent (but I know it is not easy at all). I also suggest to add LW_down and LW_up (along with TSI) as separated terms to better highlight the relative contribution of each component to the radiative balance. This is especially the case of Figure 9, but can be an idea for other pictures (Fig 4 for example).*

**Reply:** Thanks for this useful suggestion. We have modified Fig. 4 of the originally submitted manuscript (in the revised manuscript this is Fig. 5) and Fig. 9 (in the revised manuscript this refers to Fig. 10) accordingly, such that both figures include downward and upward thermal-infrared irradiances observed close to the ground, in addition to the net irradiances. Related discussions have been added to the manuscript.

[Figure]

*Figure 1: Downward (a), upward (b), and net (c) thermal-infrared irradiance simulated for averaged thermodynamic profiles observed over open ocean and sea ice during AFLUX and ACLOUD. Water vapor and clouds were switched on and off during the radiative transfer simulations.*

**Reviewer #1:** *page 2-9, suggest to replace "downward", "upward" as "downwelling" and "upwelling": downward might be interpreted as a pyranometer looking towards the surface. While for flux (which is something moving towards a direction) the terms downwelling/upwelling might be more appropriate*

**Reply:** We would rather like to keep the terminology "downward" and "upward", mostly because "downwelling" and "upwelling" indirectly refers to the electromagnetic wave-model (implied by using the term "welling") of atmospheric radiation. However, in particular for the thermal-infrared spectral range, the photon-model is often much more appropriate in the interpretation and that does not fit with the "welling" terminology. In this point we follow the terminology used by Bohren and Clothiaux (Fundamentals of Atmospheric Radiation, Wiley-VCH Verlag GmbH Weinheim, 2006). In general, that might be a matter of taste.

**Reviewer #1:** *page 2-12, intertwined -> concurrent?*

**Reply:** That has been changed accordingly.

*Reviewer #1:* page 2-27, MCAO and WAI water vapor content characteristics content can be at least qualitatively mentioned here, as water vapor (along with T) is the main responsible of F_downwelling atmospheric emission.

**Reply:** Water vapor is indeed the major atmospheric emitter, but mainly in cloud-free conditions. We appreciate this comment of reviewer #1 and have changed the corresponding sentence accordingly from:

> "In addition to local surface conditions and processes in the Arctic atmospheric boundary layer (ABL), the TNI is also determined by remote influences, such as large-scale circulation patterns that are associated, for example, with marine cold air outbreaks (MCAOs) or warm air intrusions (WAIs) (Pithan et al., 2018)."

To:

> "In addition to local surface conditions and thermodynamic profile properties, such as air temperature, water vapor content, and clouds that are the main drivers of downward atmospheric emission in the Arctic atmospheric boundary layer (ABL), the TNI is also determined by remote influences, such as large-scale circulation patterns that are associated, for example, with marine cold air outbreaks (MCAOs) or warm air intrusions (WAIs) (Pithan et al., 2018)."

*Reviewer #1:* page 4-9, kippzonen.de -> kippzonen.com

**Reply:** Changed, also on line 11.

*Reviewer #1:* page 5-1, for the KT-19 please specify the FOV and viewing nadir angle (assuming 0°).

**Reply:** We have added the following sentence to accommodate the reviewer's question:
> "The KT19 collects nadir radiance with a field of view of 2° (Ehrlich and Wendisch, 2015)."

> The following reference has been included in the list of references:
> Ehrlich, A., and M. Wendisch, 2015: Reconstruction of high-resolution time series from slow-response broadband terrestrial irradiance measurements by deconvolution. Atmos. Meas. Tech., 8, 3671-3684. www.atmos-meas-tech.net/8/3671/2015, doi:10.5194/amtd-8-3671-2015

*Reviewer #1:* page 5-26, remove the dot after "corrected.()"

**Reply:** Done

*Reviewer #1:* page 6-15, "The surface albedo and sea ice fraction are roughly linearly correlated with broader distributions...": The sea ice fraction I_f trigs the value of the surface albedo as a weighted average of open sea and ice albedo (then the 3D multiple reflections between atm-surf adjust the value). Give an idea of this in the sentence, if the authors agree.

**Reply:** We have added the following sentence to consider the comment of the reviewer:

> "The sea ice fraction modifies the value of the surface albedo as a weighted average of open ocean and sea ice albedo and the 3D multiple reflections between atmosphere and surface adjust the value."

*Reviewer #1:* page6-18, "The reason is that AFLUX was conducted earlier in the year compared to ACLOUD, and thus, during...": With such an amount of data probably the authors can investigate the SZA dependence further: $A\_clear\_sky(SZA) = A\_ocean(SZA)(1-I\_f) + A\_ice(Ts,SZA)\ I\_f$. Given $I\_f$ and $SZA, Ts$, as independent measured variables, and parametrising the $A\_ice$ and $A\_ocean$ with, for example polynomial (rather than linear) functions, it is possible to optimize the parameters providing a formulation for the albedo which can be reported and used for further studies. In cloudy conditions instead $A\_ice$ and $A\_ocean$ might be assumed constant (or dependent on surface temperature $Ts/A\_ice$).

**Reply:** The sea ice fraction considers areas covered with snow as well as bare sea ice. For snow-covered sea ice areas, the geometric thickness and liquid water content of the snow play an important role in the resulting albedo. The first factor (liquid water content of the snow) is usually approximated by the skin temperature (Tskin), as already mentioned by the reviewer. However, the dependence of surface albedo on Tskin does not follow a linear relationship (Jäkel et al., 2019). Instead, there is a small temperature range near the melting point where the surface albedo changes abruptly due to the transition from dry to wet snow. For illustration, we have plotted the data measured during AFLUX (SZA: 72° to 82°) and ACLOUD (SZA: 55° to 69°) under cloudless conditions with a sea ice coverage greater than 99 % (see Figure 2 below in this reply). The graph shows the broadband surface albedo as a function of skin temperature (Tskin) and SZA (color code). It can be seen that the variability at quasi-constant values of Tskin cannot be attributed clearly to an SZA dependence, as it is likely influenced by the structural properties of the sea ice and the snow itself. This makes it hard to come up with a useful parameterization. Anyway, since the parameterization of surface albedo is outside the scope of this study, we leave it at that and do not investigate the corresponding dependencies further.

Jäkel, E., Stapf, J., Wendisch, M., Nicolaus, M., Dorn, W., and Rinke, A.: Validation of the sea ice surface albedo scheme of the regional climate model HIRHAM-NAOSIM using aircraft measurements during the ACLOUD/PASCAL campaigns, *Cryosphere*, **13**, 1695–1708, https://doi.org/10.5194/tc-13-1695-2019, 2019.

[Figure]

*Figure 2: Broadband surface albedo measured during AFLUX and ACLOUD as a function of solar zenith angle (SZA) and skin temperature (Tskin). The data are limited to cloud-free conditions and sea ice coverage larger than 99 %.*

***Reviewer #1:*** *page 6-27, " ... increased the surface albedo" : increased the value of reflected irradiance, producing an apparent increase of the surface albedo.*

**Reply:** Done

***Reviewer #1:*** *page 10-20, The median values are mentioned here but their values not reported either here or in Figure 4. Did I missed them?*

**Reply:** These values are discussed in the manuscript. Considering the comment of the reviewer, we have added a new Table 1 to the manuscript summarizing these data:

| Season (Campaign) | Cloudy | Cloud-free |
|---|---|---|
| Winter (SHEBA) | -2 | -40 |
| Spring (AFLUX) | -6 | -60 |
| Early Summer (ACLOUD) | -11 | -75 |

**Table 1.** Examples of median values (in units of $W\ m^{-2}$) of frequency distributions of thermal-infrared net irradiances derived from different campaigns (SHEBA, AFLUX, ACLOUD) performed during different seasons (see Fig. 4).

***Reviewer #1:*** *page 10-24, "-2 to -6 Wm2, -11Wm2": are these slight variations significant considering uncertainties? Under overcast conditions and for thick clouds TNI should record approx. ~zero either in summer or winter. The shift might be due to thin cloud/non overcast? Is it possible to understand this from your data?*

**Reply:** For clouds of sufficient optical thickness in overcast conditions, the cloud emissivity $\varepsilon_{\mathrm{cld}}$ is approximately unity. Thus, the TIR downward irradiance in cloudy conditions $F^{\downarrow}_{\mathrm{tir,cld}}$ depends mostly on the cloud temperature $T_{\mathrm{cld}}$:

$$F_{\text{tir,cld}}^{\downarrow} = \sigma \cdot T_{\text{cld}}^4.$$

The emitted irradiance is reduced by the absorption of water vapor between the cloud and the surface. As indicated by the Figure 1a (see above, Fig. 7a in the revised manuscript), the effect of water vapor on $F_{\text{tir,cld}}^{\downarrow}$ in the presence of sufficiently thick clouds can be neglected. For the cloudy cases with and without considering water vapor, $F_{\text{tir,cld}}^{\downarrow}$ differs by less than 3 W m$^{-2}$ (Figure 1a above, full and striped gray bars).

Thus, it is obvious that when the cloud temperature ($T_{\text{cld}}$) and surface temperature ($T_{\text{sur}}$) are the same, the TNI should be very close to zero in overcast conditions. In situations with broken or thinner clouds, the assumption of $\varepsilon_{\text{cld}} \approx 1$ is not valid anymore, which would lead to more negative values of TNI. However, the cloudy modes indicated in Fig. 4 of the revised version of the manuscript represent clouds of sufficient optical thickness ($\varepsilon_{\text{cld}} \approx 1$), such that we can make this claim for our own data from AFLUX and ACLOUD (see Fig. 5e and 5f of the revised manuscript). Thus, we instead assume that slight temperature differences between clouds and the surface and the non-linearity of the Stefan-Boltzmann law cause the slightly negative values of the TNIs in Figs. 4 and 5 of the revised manuscript. This effect is discussed in the originally submitted manuscript version on Page 12, Lines 1–5 already. Page 12, Lines 5–9 additionally discussed the possibility of warmer cloud bases compared to the surface, leading to positive values of the TNI. Actually, this is the case in the simulations for AFLUX assuming cloudy conditions over sea ice (Figure 1c, full and striped gray bars, Fig. 7c in the revised manuscript version). While a surface cooling is derived from the simulations representing ACLOUD (-7 W m$^{-2}$), a warming of (+5 W m$^{-2}$) is simulated for AFLUX.

Although the differences between the median values of the thermal-infrared net irradiances (TNIs) for cloudy conditions are small, we consider them significant taking into account the large amount of data collected using the identical instrumentation reducing systematic uncertainties in the net irradiances.

***Reviewer #1:*** *page 10-26, "These strongly ...: How the surface low RH or, better, IWV content, were excluded as a primary cause of these extreme negative values? Relatively dry air advection causes a low incoming irradiance as the emissivity of the clear-sky depends on the WV content mostly. It would also be worth to include a picture supporting your conclusions, showing nilas (hemy camera) and high surface temperature T (KT-19) and F_up (pyrgeometer).*

To support this claim, we have plotted in Figure 3 (see below) the probability density functions (pdfs) of the observed downward (left panel), upward (center panel), and net (right panel) thermal-infrared irradiances including all cases (red line, cloud-free plus cloudy), all cloud-free cases (full red, excluding the measurements in cloudy conditions), and the cases with very low TNI values (blue line, defined by $F_{\text{tir,net}}$ < -80 W m$^{-2}$). For the downward irradiances (left panel) no differences between all cloud-free and low TNI pdfs are observed. However, for the upward irradiances (center panel) a significant difference between the two pdfs is obvious, which leads to very low TNIs, also shown in the right panel. This is solid evidence that the upward irradiance driving surface emitted thermal-infrared irradiances determines the very low TNI values, and it seems conclusive that these large values of emitted upward thermal-infrared irradiances are emitted sporadically by high-temperature surfaces such as nilas or thin ice flows.

[Figure]

Figure 3: Probability density functions (pdfs) of the observed downward (left panel), upward (center panel), and net (right panel) thermal-infrared irradiances including all cases (red line, cloud-free and cloudy), all cloud-free cases (full red, excluding the measurements in cloudy conditions), and the cases with very low TNI values (blue line, $F_{tir,net} < -80\ W\ m^{-2}$).

**Reviewer #1:** *Figure 4, Suggestion: Despite the dotted/dashed, blue and black appears too similar. Try using a light color for the blue or a green for the black. Distinguish N-ICE with solid/dashed rather than dotted/dashed.*

**Reply:** We have used green for the blue-dashed (N-ICE2015), and kept the black for SHEBA. Then we have changed the blue-dotted to blue solid to better distinguish N-ICE. We think that this choice enables best to distinguish the lines. Thanks for pointing this out.

**Reviewer #1:** *page 11-eq.2 (and line 5-10), This is a major concern. F_incoming ~ \eps \sigma T_atm^4?; what about the apparent emissivity \eps (which should probably range around 0.5-0.6) in this equation? See for example Busetto et al., 2013 (Antarctic Science). The value used for T_atm depends on the aircraft level. It is worth to use a stable value at the top of the boundary layer to represent the thermodynamic status of the atmosphere, coupled with a suitable function for the apparent emissivity. The above reference report a number of reference to historical parametrization. Hence, this reviewer is not sure to agree on the discussion given below as the equation should appear as F_down ~ eps(T,rh,...) sigma T_atm^4 with a T_atm that can be derived from the dropsondes combined with aircraft records. Did the authors really verified their eq. 2 with data collected in field?*

**Reply:** The reviewer is correct that, according to the Stefan-Boltzmann law, the irradiance emitted by a medium of emissivity $\varepsilon$ is calculated by:

$$F = \varepsilon \cdot \sigma \cdot T^4,$$

where $T$ is the temperature of the medium. $\varepsilon$ depends, e.g., on the atmospheric water vapor content.

More precisely, the spectral diffuse radiance $I_{\text{diff},\lambda}$ along a slant path $s$ is calculated as follows ($\tau$ being the optical thickness):

$$I_{\text{diff},\lambda}\left[\tau(s) = 0\right] = I_{\text{diff},\lambda}\left[\tau'(s)\right] \cdot e^{-\tau'(s)} + \int_0^{\tau'(s)} B_\lambda\left\{T[\tau(s)]\right\} \cdot e^{-\tau(s)}\ d\tau(s)$$

The first term on the right side of this equation represents the attenuation of the diffuse spectral radiance, the second term includes emission and absorption of photons along their way to the receiver. If we apply this equation to a vertical path, then the slant path $s$ is replaced by altitude $z$.

Since the atmospheric temperature and humidity profiles are not constant with height $z$, $T(z)$ and $\varepsilon(z)$ depend on the atmospheric layer. A part of the irradiance emitted from a specific layer towards the surface is attenuated along its way.

However, in this part of the manuscript we just roughly estimate irradiance components and assume the atmosphere as a bulk layer. Therefore, we refer to $T_{\mathrm{atm}}$ as the **effective brightness temperature** of the atmosphere, which results from the combination of the temperature and humidity profiles, and which implicitly includes the apparent (effective) emissivity the reviewer refers to. Thus, the TIR downward irradiance in cloud-free conditions $F_{\mathrm{tir,cf}}^{\downarrow}$ is given by:

$$F_{\mathrm{tir,cf}}^{\downarrow} = \sigma \cdot T_{\mathrm{atm}}^4.$$

For the surface, an emissivity $\varepsilon_{\mathrm{sur}} = 1$ is assumed, which almost agrees with the emissivity of ocean water or sea ice/snow. Thus, the surface brightness temperature $T_{\mathrm{sur}}$ is almost identical to the physical surface temperature, and the TIR upward irradiance $F_{\mathrm{tir,cf}}^{\uparrow}$ is given by:

$$F_{\mathrm{tir,cf}}^{\uparrow} = \sigma \cdot T_{\mathrm{sur}}^4,$$

Thus, when considering the atmospheric effective brightness temperature, the simple and rather qualitative discussion given on Page 11, Lines 5–10 makes sense and can be corroborated by results from the radiative transfer simulations.

The simulated cloud-free TNI ($F_{\mathrm{tir,net}}$) (Figure 1c above, corresponds to Fig. 7c in the revised manuscript) over sea ice including water vapor (full yellow bar) shows a larger surface cooling during ACLOUD (–82 W m$^{-2}$) compared to AFLUX (–72 W m$^{-2}$). This is in accordance with Fig. 4 of the manuscript (Figs. 5e and 5f in the revised version), despite the small differences between the observed (Figs. 5e and 5f in the revised version) and simulated (Fig. 7c in the revised manuscript) values. $F_{\mathrm{tir}}^{\downarrow}$ (Figure 7a) increases from 156 W m$^{-2}$ during AFLUX to 223 W m$^{-2}$ during ACLOUD (difference of 67 W m$^{-2}$). $F_{\mathrm{tir}}^{\uparrow}$ (Figure 7b) shows a larger increase of 78 W m$^{-2}$ (from 228 W m$^{-2}$ during AFLUX to 306 W m$^{-2}$ during ACLOUD) due to the higher atmosphere/surface temperatures ($F \sim T^4$). This implies that, due to the non-linearity of the Stefan-Boltzmann law, the change of the surface temperature has the larger effect on the TNI variability between both campaigns.

The individual contributions of the temperature and humidity profiles on $F_{\mathrm{tir}}^{\downarrow}$ are discussed in the answer on comment *page 12-10* (below).

*Reviewer #1: page 12-10, This is a major concern (given here but as a comment to the work as a whole). "various processes": The explanation of the values obtained through a basic statistical analysis of summer and spring periods are attributed quite arbitrarily to the combination of possible processes. It is difficult to discern between the various contribution as a multi-variate analysis approach was not conducted (if feasible). The arguments given in the discussion are all plausible, and the author proven to well known the processes involved, but one of the objective of this research should be to identify, for each phenomena (cloud status, atm thermodinamic status, air advection, surface properties, ...), its relative contribution. This aspect should be reinforced possibly using a robust mathematical parametrizations.*

**Reply:** In general, this is a very interesting aspect, but far beyond the scope and our possibilities for this paper. Realizing this idea would require a completely different paper including much more modeling where we could play with different scenarios. Parameterizations of the suggested type are simply not existing.

The reasons for this specific variability (increase of spread between cloudy and cloud-free TNI from winter to summer) given on Page 12, Line 10 was basically discussed in the answers of the previous two comments and is also given in the manuscript (the brightness temperature difference between $T_{\mathrm{atm}}/T_{\mathrm{cld}}$ and $T_{\mathrm{sur}}$ is larger and, thus, the non-linearity of the Stefan-Boltzmann law plays a larger role in cloudy than in cloud-free conditions leading to a more variable $F_{\mathrm{tir}}^{\downarrow}$ in cloudy compared to cloud-free conditions).

However, we understand that this comment is intended to be more general rather than related to only this specific sentence. Unfortunately, the separation of the temperature and humidity effects on $F_{\mathrm{tir}}^{\downarrow}$ is not trivial, because temperature and the absolute humidity $\rho_{\mathrm{wv}}$, which determine the radiative emissivity, are not independent of each other. With increasing temperature, $\rho_{\mathrm{wv}}$ tends to increase due to the exponential Clausius-Clapeyron equation (increasing equilibrium water vapor pressure with increasing temperature). Both increases contribute to the increase of $F_{\mathrm{tir}}^{\downarrow}$. We applied different approaches to separate both contributions.

First, the water vapor is excluded entirely from the radiative transfer simulations to isolate the effect of the temperature profile on $F_{\mathrm{tir}}^{\downarrow}$. In this case, the emission/absorption profile is determined by the remaining atmospheric gases. In cloudy conditions, the independence of $F_{\mathrm{tir}}^{\downarrow}$ on the water vapor was discussed above. In cloud-free conditions, the resulting $F_{\mathrm{tir}}^{\downarrow}$ over sea ice (open ocean) is 60 W m$^{-2}$ (66 W m$^{-2}$) during AFLUX and 82 W m$^{-2}$ (85 W m$^{-2}$) during ACLOUD. The difference to the simulations including water vapor would then quantify the water vapor effect on $F_{\mathrm{tir}}^{\downarrow}$. It amounts to 96 W m$^{-2}$ (113 W m$^{-2}$) during AFLUX and 141 W m$^{-2}$ (153 W m$^{-2}$) during ACLOUD. Obviously, the water vapor has a larger contribution on $F_{\mathrm{tir}}^{\downarrow}$, which ranges between 62 % and 64 %. This means that the water vapor increases the effective atmospheric emissivity by a factor of approximately 2.7 in our cases. In absolute numbers, however, the water vapor contribution to $F_{\mathrm{tir}}^{\downarrow}$ is not independent of temperature. Also the variability of cloud-free of $F_{\mathrm{tir,cf}}^{\downarrow}$ between the surface types and the campaigns is dominated by the water vapor effect.

Especially regarding seasonal variability, the temperature-induced change of $\rho_{\mathrm{wv}}$ raises the question of whether this change forms part of the temperature effect or represents a separate water vapor effect on $F_{\mathrm{tir}}^{\downarrow}$. We examine both variants with two additional approaches. Another set of simulations is performed using the temperature profiles of each campaign–surface type combination and combining them with humidity profiles of the other campaign or surface type. These simulations should isolate the pure humidity effect on the variability of $F_{\mathrm{tir}}^{\downarrow}$.

A temperature-induced change of $\rho_{\mathrm{wv}}$ does not change the relative humidity (*RH*). If this change should be a part of the temperature effect, the real water vapor effect results from a variation of the *RH* only. Thus, we first vary the *RH* profiles in our additional simulations. Figure 4 (see below) shows the *RH* profiles of all situations. Since *RH* does not differ significantly between the campaigns/surface types, the variation of the *RH* profiles causes a change of $F_{\mathrm{tir}}^{\downarrow}$ of no more than 3 W m$^{-2}$ (not shown), and typically much less. Thus, using this approach, almost the entire contribution to the variability of $F_{\mathrm{tir}}^{\downarrow}$ results from the temperature change.

Next, we vary the $\rho_{\mathrm{wv}}$ profiles in our simulations such that the temperature-induced change of $\rho_{\mathrm{wv}}$ also contributes to the water vapor effect. These combinations partly result in unrealistically large

super-saturations with *RH* of up to 250 %. Nevertheless, the resulting irradiances are indicated by symbols in Figure 1 (see above). Due to the insensitivity of $F_{tir}^{\downarrow}$ to water vapor in cloudy conditions, the variation of the $\rho_{wv}$ profile changes $F_{tir}^{\downarrow}$ by less than 1.5 W m$^{-2}$. In cloud-free conditions, the difference between $F_{tir}^{\downarrow}$ over sea ice and open ocean during AFLUX is 22 W m$^{-2}$. However, using the $\rho_{wv}$ profile over open ocean together with the temperature profile over sea ice increases $F_{tir}^{\downarrow}$ by only 3 W m$^{-2}$ compared to using both sea ice profiles. Similarly, $F_{tir}^{\downarrow}$ increases by 66 W m$^{-2}$ from AFLUX to ACLOUD over sea ice, while only changing the $\rho_{wv}$ profile to ACLOUD conditions accounts for only 15 W m$^{-2}$. These numbers indicate that the water vapor effect is relatively less influential for the variability of $F_{tir}^{\downarrow}$.

In summary, our analyses showed that the contribution of the water vapor to $F_{tir}^{\downarrow}$ depends on the approach of how to disentangle the temperature and humidity effects. In our case, the water vapor is dominant when its contribution is simply defined by its presence. In contrast, the impact of water vapor on the variability of $F_{tir}^{\downarrow}$ is weaker or nearly absent, depending on whether the change of $\rho_{wv}$ due to a temperature change is considered as a separate water vapor effect or not.

[Figure]

*Figure 4: Average RH profiles observed by dropsondes during AFLUX (a) and ACLOUD (b) above sea ice (solid lines) and open ocean (dashed lines, all-sky conditions). Sea ice and open ocean areas are defined as regions with a sea ice concentration above 90 % and below 10 %, respectively, derived from daily sea ice concentrations maps from Spreen et al. (2008).*

**Reviewer #1:** *page 13-3, "It can be also interpreted as a ..." -> "It is a ..."*

**Reply:** Done.

**Reviewer #1:** *page 13-10, "relatively invariant cloud base height and temperature":  give median and interquantile range values or something supporting this argument*

**Reply:** To show this, we have copied Figure 5 (revised manuscript) below and additionally included median and interquartile ranges (IQR, black bar). We have done so for the cloudy modes, defined as LWP > 5 g m$^{-2}$. Obviously, the IQRs for AFLUX are larger than for ACLOUD, in particular over sea ice. That supports our claim of more variable thermodynamic conditions during AFLUX, although we have no statistical significant data of the cloud base height and temperature. The insignificant humidity

influence in cloudy conditions on downward, upward, and net thermal-infrared irradiances shown above corroborates this conclusion.

To quantify the variabilities, have a look at the numbers given in Table 2 of the revised manuscript copied below.

[Figure]

*Figure 5: Median (full dots) and interquartile ranges (IQR, black bars) included into Figure 5 of the revised manuscript).*

| | $F_{\text{tir}}^{\downarrow}$ | [W m$^{-2}$] | | $F_{\text{tir}}^{\uparrow}$ | [W m$^{-2}$] | | $F_{\text{tir,net}}$ | [W m$^{-2}$] | |
|---|---|---|---|---|---|---|---|---|---|
| | Median | 25pct | 75pct | Median | 25pct | 75pct | Median | 25pct | 75pct |
| AFLUX (open ocean) | 239 | 224 | 251 | 292 | 286 | 297 | -52 | -68 | -43 |
| AFLUX (sea ice) | 215 | 198 | 251 | 244 | 219 | 257 | -12 | -37 | -2 |
| ACLOUD (open ocean) | 302 | 290 | 308 | 323 | 313 | 331 | -24 | -31 | -15 |
| ACLOUD (sea ice) | 295 | 285 | 304 | 311 | 305 | 314 | -13 | -19 | -10 |

**Table 2.** Median, and percentile (pct, 25 % and 75%) values (in units of W m$^{-2}$) of the cloudy modes (defined as LWP > 5 g m$^{-2}$) of the downward, upward, and net thermal-infrared irradiances ($F_{\text{tir}}^{\downarrow}$, $F_{\text{tir}}^{\uparrow}$, and $F_{\text{tir,net}}$) measured during low-level flights during AFLUX and ACLOUD, as observed over different surface types (open ocean and sea ice) (see Fig. 5).

***Reviewer #1:*** *page 13-10, "more data are available for ACLOUD" -> quantify (number of flights? matrix data dimension?)*

**Reply:** During ACLOUD we have collected 16 hours, during AFLUX 6 hours of data during low-level flights (average flight altitude between 70 m and 80 m). This info is already given in Section 2. We implemented this info in compressed form on page 13-10, as requested by the reviewer.

*Reviewer #1: page 13-13:14, speculative in my opinion. It could, yes, but it is not proven by the observation/analysis presented here. Suggest to remove.*

**Reply:** We have deleted the corresponding part of the sentence.

*Reviewer #1: page 14-5, remove "should"*

**Reply:** Done.

*Reviewer #1: page 14-16, "..., which mostly considers the solar spectral range that is not directly relevant for the TNI" --> "..., as BT determines the upwelling component of the thermal radiation budget."*

**Reply:** We have changed the corresponding part of the sentence from:
"…, which mostly considers the solar spectral range that is not directly relevant for the TNI."
To
"…, which mostly considers the solar spectral range that is not directly relevant for the TNI as BT determines the upward component of the thermal-infrared radiation budget."

*Reviewer #1: page 15-Figure 6 caption remarks that the x-axes of panel (b) is reversed to highlight the position of the modes in the two figures, supporting page 14-lines 8:10 discussion.*

**Reply:** We have added the following sentence to the figure caption of Fig. 6:
"The x-axis of panel (b) is reversed to highlight the position of the modes similar to Fig. 5."

*Reviewer #1: page 15-4, "that are caused during transformation processes that air masses experience during meridional transport in more detail in this section" --> "that are caused by meridional transport of air masses with different thermodynamic properties in more detail in this section"*

**Reply:** An air mass is, by definition, characterized by mostly homogeneous properties. They change during their transport over inhomogeneous ground. Thus, we would prefer not to change this sentence.

*Reviewer #1: page 16-15, The simulations can also be validated w.r.t. the ground based data from the AWIPEV station BSRN (Ny Alesund, ask Maturilli/Driemel, see NYa station on BSRN site bsrn.awi.de), and/or mauro.mazzola@cnr.it for data from the Amudsen Nobile Climate Change tower (https://bo.isp.cnr.it/main/CCTower/?Home)*

**Reply:** We have refrained from doing so because we were not flying over Ny-Ålesund all the time.

***Reviewer #1:*** *page 16-30, (remark-remark-remark moisture effect) Is it possible, from data at disposal, to quantify the effect of moisture in more details?*

**Reply:** We did so by including Figure 7 and corresponding discussion into the revised manuscript.

***Reviewer #1:*** *page 17-6, "air mass transformation" --> "air mass difference/heterogeneity" ? Transformation might take longer periods to occur. Air mass flow over the area of interest with its heterogeneity.*

**Reply:** We don't see an issue using the term air mass transformation here.

***Reviewer #1:*** *page 18-1, same here, not sure about the balance among the mixing air massess or just a push off the cold air by the advection of warm air. Likely both (purely philosophical/definition comment probably).*

**Reply:** See discussion above.

***Reviewer #1:*** *page 19-Figure 8, Add Ny-Alesund to the map.*

**Reply:** Done. We have deleted Longyearbyen.

***Reviewer #1:*** *page 21-29:33, To support this, I suggest to report downwelling and upwelling LW measurements/simulations in Figure 9 (adding panels?). Not really sure about the usefulness of the clear-sky simulations here but they might serve as a baseline for the discussion about mutual cancellation of surface effects and thermodynamic states between open ocean and sea ice.*

**Reply:** Done. The cloud-free simulation results have been deleted from the figure.

***Reviewer #1:*** *Figure 9: add Ny Alesund to the map if it fits, ... then use a different symbols in panel (a) for each day considered, so that one can easier associate it to panel (b).*

**Reply:** Done.

***Reviewer #1:*** *page 23-23:25, possibly, but a bit speculative.*

**Reply:** We have now convincingly shown this above.

*Reviewer #1: page 23-26, Processes that play ...: enumerate/mention the processes the authors have in mind, to avoid generalization ... page 23-28, Are "radiative transfer and thermodynamic processes" included in the hidden list of the previous sentence?*

**Reply:** Yes, indeed, we have modified this sentence accordingly.

---

## Author Comment (AC2)

**Response to Reviewer #2**

Thank you for your comprehensive and valuable comments and suggestions, which we have carefully considered. We very much appreciate your thoughts and time; your ideas really improved and enriched the manuscript! Please look at our responses and the revised manuscript.

The text of your review is copied below in *Calibri black font italic style*; our replies are given in black style.

*Reviewer #2:*

*An extensive analysis of low-level airborne observations of near-surface thermal-infrared net irradiance (TNI) conducted in the Arctic is presented in the article "Effects of variable, ice-ocean surface properties and air mass transformation on the Arctic radiative energy budget". The study explores the complex interplay between sea ice concentration, cloud properties, surface albedo and atmospheric thermodynamics, shedding light on the radiative transfer and thermodynamic processes occurring in the marginal ice zone (MIZ) and surrounding areas. The article addresses the critical knowledge gap regarding the understanding of near-surface thermal-infrared net irradiance in the Arctic, particularly in the MIZ. Several noteworthy findings are presented. Firstly, the study reveals four distinct modes of near-surface TNI in the MIZ. Furthermore, the authors observe a seasonal shift in the TNI field above homogeneous sea ice, characterized by more negative net irradiances and increased occurrence of cloudy states in summer compared to winter. This shift is attributed to the non-linearity of the Planck emission. Moreover, the authors highlight the importance of considering the thermodynamic profiles and stability in both cloudy and cloud-free conditions, as these factors significantly impact the TNI. The amount and quality of data evaluated in chapter 4 ("Cloudy and cloud-free modes of thermal-infrared net irradiance (TNI)") is impressive and the conclusions comprehensible and interesting.*

*Naturally, there is much less data for the analysis in chapter 5 ("Impact of synoptically driven meridional air mass transports on thermal-infrared radiation at the surface") and the conclusions on the individual case studies are more speculative here. While the publication mentions diverging and variable estimates from satellites, models, and observations, there is limited comparison or discussion of the findings in relation to existing literature. How could more robust statements be made here in the future in order to get similar TNI values from remote sensing, models and in situ observations?*

*The research presented in this article carries significant implications for our understanding of radiative transfer and thermodynamic processes in the Arctic. The findings provide valuable insights into the complexities of TNI in the MIZ and its relationship with sea ice dynamics. Publication is recommended and addressing the above-mentioned minor points of criticism would further strengthen the study.*

**Reply:**

The first part of the reviewer's comment is about placing the findings in relation to existing literature. The paper has attempted to do so by mentioning some of the key observations over sea ice from the past, such as from SHEBA and N-ICE2015, as well as incorporating some data from those campaigns into the analysis as direct context for the ACLOUD and AFLUX measurements. There are relatively few of these measurements that can be used as context. The MOSAiC data are one potential new source, although analysis of that data is still underway and will be published in the future.

The second part of the reviewer's comment specifically asked if more robust statements could be made in the future to get similar TNI values from different perspectives. We interpret this statement as asking how we can better achieve consistency across the different perspectives on TNI. Indeed, this is an important goal and one of the motivations for making the aircraft-based measurements that are the subject of this paper. These observations are relatively unique and offer good potential to evaluate how both models and satellite retrievals represent the TNI across the ocean-ice interface transition zone, including their representation of the different states of the TNI distribution. Achieving this consistency is a long-term process that will require further observations, detailed intercomparisons, more sophisticated model-process diagnostics, and further refinement of satellite retrieval approaches as well as model parameterizations. While that work is surely beyond the scope of this paper, we have added some statements to the last paragraph of the paper that express the need for this important future research.

***Reviewer #2:*** *Figure 8 change F_ter,net --> F_tin*

**Reply:** Thanks for carefully reading the text, we have changed $F_{\text{ter,net}}$ to $F_{\text{tir,net}}$

***Reviewer #2:*** *Figure 9 – seems a bit cluttered and confusing to me. Showing simulated thermal-infrared downward irradiances for clear sky conditions adds little value.*

**Reply:** The cloud-free values along the flight path are used to obtain an appropriate baseline. However, we agree with the comment of the reviewer, which was also made by reviewer #1. Therefore, we have skipped the results of the simulations for cloud-free conditions. Instead, we have added two panels of downward and upward irradiances to this figure, which simplifies interpretation.